# Distribution Learning with Valid Outputs Beyond the Worst-Case

**Nick Rittler**
University of California - San Diego
nrittler@ucsd.edu

**Kamalika Chaudhuri**
University of California - San Diego
kamalika@cs.ucsd.edu

## Abstract

Generative models at times produce "invalid" outputs, such as images with generation artifacts and unnatural sounds. Validity-constrained distribution learning attempts to address this problem by requiring that the learned distribution have a provably small fraction of its mass in invalid parts of space – something which standard loss minimization does not always ensure. To this end, a learner in this model can guide the learning via "validity queries", which allow it to ascertain the validity of individual examples. Prior work on this problem takes a worst-case stance, showing that proper learning requires an exponential number of validity queries, and demonstrating an improper algorithm which – while generating guarantees in a wide-range of settings – makes an atypical polynomial number of validity queries. In this work, we take a first step towards characterizing regimes where guaranteeing validity is easier than in the worst-case. We show that when the data distribution lies in the model class and the log-loss is minimized, the number of samples required to ensure validity has a weak dependence on the validity requirement. Additionally, we show that when the validity region belongs to a VC-class, a limited number of validity queries are often sufficient.

## 1 Introduction

When sampling from a generative model, it is highly desirable that its outputs meet some basic criteria of quality. In the case of text, this may mean that generated sentences respect grammar rules, or avoid the use of biased or offensive language [1, 2]. When generating code, a criterion may be that the generated code successfully compiles [3]. In image generation, we might wish to avoid blurry outputs, or those possessing generation artifacts which clearly distinguish them from natural images [4, 5].

In this paper, we examine the statistical cost of ensuring that learned distributions produce such "valid" outputs. To do so, we consider an elegant formulation of the problem of learning such valid models due to [3]. In their work, training data are generated according to a probability distribution $P$, and the binary "validity" of examples is determined by some unknown "validity function" $v$. Given sample access to $P$ and query access to $v$, a learner attempts to identify a probability distribution which outputs invalid examples with probability at most $\epsilon_2$. At the same time, the distribution should have a loss which is at most $\epsilon_1$ worse than that of the minimum loss model in a class $\mathcal{Q}$ which outputs valid examples with probability 1. Here, query access to $v$ captures the idea that collecting samples is often cheap, but verifying validity is often less so, possibly requiring a human-in-the-loop.

The initial work of [3] suggests that choosing such a low-loss, high-validity distribution $\hat{q}$ may require a large number of validity queries. Under the assumption that $P$ is "fully-valid", i.e. outputs a valid example with probability 1, they show that in the worst case, $2^{\Omega(1/\epsilon_1)}$ validity queries are required to choose such a model $\hat{q}$ from the class $\mathcal{Q}$. They follow this result with an improper learning algorithm

for choosing $\hat{q}$ which, while achieving polynomial bounds on the number of validity queries, uses a relatively large number of validity queries $\tilde{O}\left(\log(|\mathcal{Q}|)/\epsilon_1^2\epsilon_2\right)$.

The somewhat pessimistic picture painted by these fascinating complexity-theoretic results can be tracked to their generality. Firstly, it's possible that $\mathcal{Q}$ and $P$ are significantly "mismatched", i.e. the support of each model $q \in \mathcal{Q}$ has only a small overlap with the support of the distribution $P$, in which case the validity information contained in valid training samples is unhelpful to a proper learner. Secondly, their improper learning algorithm is largely loss-agnostic, in that it generates guarantees for a wide class of bounded loss functions. Finally, nothing is assumed about the form of the validity function $v$, precluding provable estimation.

In this work, we offer a counterbalance to this picture, beginning an investigation into learning settings where guaranteeing validity is cheaper than such results might indicate. We first consider learning under complete elimination of model class mismatch, where $\mathcal{Q}$ is rich enough to contain the fully-valid data distribution $P$, and the loss is the log-loss $l(f(x)) = \log(1/f(x))$. It is intuitive that in this setting, loss minimization alone should guarantee validity. Somewhat less intuitively, we demonstrate an algorithm closely related to empirical risk minimization which uses just $\tilde{O}\left(\log(|\mathcal{Q}|)/\min(\epsilon_1^2, \epsilon_2)\right)$ samples to guarantee its output meets loss and validity requirements – in other words, validity comes quickly under random sampling from $P$ in this setting.

Secondly, we consider learning under a different realizability assumption, namely that the validity region is a member of a VC-class of dimension $D$. In this setting, we provide an analysis of the natural scheme of restricting the empirical risk minimizer to an estimate of the valid part of space. We show that when small-loss models $q \in \mathcal{Q}$ have at least constant validity, this scheme uses $\tilde{O}\left(D/\epsilon_2\right)$ validity queries, implying a query cost reduction over the general-purpose algorithm of [3]. We also show that learning under the capped log-loss can be used to relax the assumption of constant validity at the cost of an extra factor of $1/\epsilon_1$.

Our results suggest the existence of a rich web of settings in which validity may be cheaper than in the general case. They also suggest that the choice of the loss plays an important roll in guaranteeing valid outputs, compelling further investigation of the log-loss in particular.

## 2  Related Work

The framing of learning distributiuons in terms of PAC guarantees similar to [6] dates back to [7], who consider the learnability of specific classes of discrete distributions under a realizability assumption. A significant body of work on distribution learning has been developed overtime, often focusing on algorithms for learning over parametric families or under specific "structural" assumptions [8, 9, 10, 11]. The only theoretical contribution to validity-constrained distribution learning under the formulation posed by [3] that we are aware of is that work itself.

The study of loss functions for the evaluation of probabilistic models has often been studied the lens of "scoring rules" in the forecasting literature [12, 13, 14]. There are some notable recent contributions towards expanding the understanding of when loss functions for distribution learning display desirable properties, e.g. "properness", which designates that the loss is minimized by the true data distribution [8, 15].

The first half of this paper draws on intuition from hypothesis testing to evaluate the performance of empirical risk minimization. Hypothesis testing is a major focus of the classical statistics literature [16]. The bounds in the first half of the paper are due to analysis inspired by the Neyman-Pearson lemma [17, 18], and rely on the approximation of total variational distance between product measures [19].

The applied literature on generative modeling has consistently noted the problem of learned models producing "invalid" examples [20, 21, 22, 4]. Various techniques have been proposed for mitigating invalidity generally, and in domain specific settings [23, 20, 24]. While working under the assumption that the validity function lies in a VC-class, the strategy we introduce has some rough semblance to a "post-editing" procedure proposed by [24].

# 3 Preliminaries

## 3.1 Problem Setup

Let $\mathcal{X}$ be a subset of Euclidean space $\mathbb{R}^d$ with finite Lebesgue measure $\lambda$. Let $\mathcal{P}$ denote the set of all probability distributions on the measurable space $(\mathcal{X}, \mathcal{F}_{\mathcal{X}})$, where $\mathcal{F}_{\mathcal{X}}$ arises from Lebesgue measurable sets intersected with $\mathcal{X}$. Let $P \in \mathcal{P}$ be the data-generating distribution.

In the eyes of the learner, the function $v : \mathcal{X} \to \{0, 1\}$ is a fixed and unknown "validity function", measurable with respect to the relevant distributions. The validity function denotes whether or not an example $x \in \mathcal{X}$ is considered a valid output for a learned approximation of $P$. The learner is given a model class of $\mathcal{Q} \subset \mathcal{P}$ of probability distributions on $\mathcal{X}$, each with density $f_q$ with respect to $\lambda$, and afforded with the knowledge that at least one $q \in \mathcal{Q}$ is "fully-valid", i.e. that there is some $q \in \mathcal{Q}$ with invalidity $V(q) := \Pr_{X \sim q}(v(X) = 1) = 1$. We at times use the notion of "invalidity" of a model, by which we mean $I(q) = 1 - V(q)$. Following the main exposition of [3], we assume $\mathcal{Q}$ is of finite cardinality.

The goodness-of-fit of a model $q \in \mathcal{P}$ is governed by a decreasing "local" loss function $l : \mathbb{R}^{\geq 0} \to \mathbb{R} \cup \{\infty\}$. Such a loss function gives rise to loss of model via $L_P(q; l) := \mathbb{E}_{X \sim P}[l(f_q(X))]$. Given an i.i.d sample $S$ from $P$, we let the empirical estimate of the loss of a model be $L_S(q; l) = \sum_{x_i \in S} l(f_q(x_i))/|S|$. We use the shorthand $L_P(q)$ and $L_S(q)$ to denote the true and empirical losses of $q$ under the log-loss $l(q) = \log(1/f_q(x))$, where $\log$ denotes the natural logarithm. We take the log-loss to be infinite at points where $f_q(x) = 0$.

## 3.2 Goal of Learning

The goal of the learner is to choose some $\hat{q} \in \mathcal{P}$ which has a loss $L_P(\hat{q}; l)$ similar to that of the lowest-loss fully-valid model in $\mathcal{Q}$, while simultaneously maintaining near full-validity. Explicitly, consider the model

$$q^* := \underset{q \in \mathcal{Q}: V(q) = 1}{\arg\min} \; L_P(q; l).$$

To describe the quality of an outputted model, we consider two learning parameters $\epsilon_1$ and $\epsilon_2$, where $\epsilon_1$ is used to control the loss sub-optimality, and $\epsilon_2$ to control the invalidity. Formally then, the goal of the learner is to output $\hat{q} \in \mathcal{P}$ satisfying $L(\hat{q}) \leq L(q^*) + \epsilon_1$ and $I(\hat{q}) \leq \epsilon_2$. To accomplish this goal, the learner has sample access to $P$, and query access to $v$, i.e. a learner can draw any finite number of $i.i.d.$ samples from $P$, and any request the value of the validity function $v$ at any finite number of inputs in $\mathcal{X}$.

At a minimum, we are interested in algorithms which require a number of samples from $P$ and number validity queries that is polynomial in $\log(|\mathcal{Q}|)$, $1/\epsilon_1$ and $1/\epsilon_2$. Ideally, we would like to minimize the number of validity queries given some polynomial number of samples from $P$. The motivation for this goal is similar to the minimization of label queries in active learning for classification [25], where samples from the marginal over instances are often cheap, but labeling such examples is assumed expensive.

## 3.3 Full-Validity of $P$

We assume that all samples from the data-generating distribution $P$ are valid, i.e. that $V(P) = 1$. Under such an assumption, the query demand of a learning algorithm can be conceptualized as the overhead number of queries sufficient for choosing a good model under the standard procedure of removing invalid examples from the training set.

If the data distribution is not fully-valid, and valid samples are required by an algorithm, the question of minimizing the overall number of queries is dependent on the sample complexity of learning – if one assumes that $P$ has been constructed by "accepting" valid samples from some underlying distribution which outputs a valid sample with constant probability, then the overall query cost incurred by an algorithm is on the order of the larger of the number of samples and the number of "overhead" validity queries it uses.

In this paper, we are primarily interested in the "overhead" number of queries, which we refer to as the "number of validity queries" of a given scheme. In most cases, algorithm sample requirements are similar to $O(\log(|\mathcal{Q}|)/\epsilon_1^2)$, which allows for accurate loss estimation in many settings.

### 3.4 Summary of Previous Results

The learning problem above is due to [3], who considered the possibility of specifying learning algorithms meeting the above bi-criteria objective for any choice of bounded, decreasing, local loss function.

This work gives some interesting insight into the difficulty of selecting such a low-loss, high-validity model. They begin by giving a negative result, namely that any proper learning algorithm outputting $\hat{q} \in \mathcal{Q}$, must make $2^{\Omega(1/\epsilon_1)}$ validity queries in the worst case, regardless of the number of samples available from $P$. This result arises from a specific problem instance wherein every $q \in \mathcal{Q}$ has a significant amount of mass outside of the support of $P$, in which case samples from a fully-valid $P$ do not give information about $v$ in parts of space relevant to the choice of $\hat{q} \in \mathcal{Q}$.

On the other hand, they demonstrate an improper learning algorithm which achieves polynomial bounds on samples and validity queries for any choice of loss meeting the above criteria. Their algorithm harnesses a constrained ERM oracle, iteratively querying the validity of samples from the model $q \in \mathcal{Q}$ which is the empirical loss minimizer putting no mass on points known to be invalid. In particular, their scheme uses $\tilde{O}(\log(|\mathcal{Q}|)/\epsilon_1^2)$ samples and $\tilde{O}(\log(|\mathcal{Q}|)/\epsilon_1^2\epsilon_2)$ validity queries.

## 4 Learning Without Model Class Mismatch Under the Log-Loss

We first consider the problem of selecting a low-loss, high-validity model under a relaxation of two of the main sources of difficulty in original problem formulation: the misalignment of the model class $\mathcal{Q}$ with the data distribution $P$, and the lack of assumptions on the loss.

In particular, we consider the problem under a realizability assumption, namely that $P \in \mathcal{Q}$, further investigating the power of the log-loss. Such a setting is arguably more closely aligned with contemporary learning settings with rich model classes that appropriately capture features of the underlying data distribution, where the validity information contained in samples from $P$ can be exploited by convergence to the best information-theoretic representation of $P$ in $\mathcal{Q}$.

The log-loss is by far the most widely-used loss in practice [8]. It is a classic result of the proper scoring rule literature that the log-loss is the unique strictly-proper local loss, i.e. the only local loss under which for all distributions $q \neq P$, it holds that $L_P(P;l) < L_P(q;l)$. This highly desirable property – implying that convergence to the optimum over $\mathcal{P}$ coincides with convergence to $P$ – makes the choice of an alternative outside of capped variants preferable only under specialized circumstances.

### 4.1 Towards Validity without Validity Queries

Given that samples are assumed to be valid, and the log-loss permits convergence to the data generating distribution, one would hope that simply selecting a model $\hat{q} \in \mathcal{Q}$ which is a sufficiently good representation of $P$ under the log-loss would yield validity guarantees in this setting. Simply utilizing empirical risk minimization (ERM) is the canonical approach to this end, and one which, given sufficient data from $P$, uses exactly zero validity queries.

Note that any model $q$ with invalidity $I(q) > \epsilon_2$ necessary has $d_{TV}(q, P) > \epsilon_2$. In this case, $q$ must have at least $\epsilon_2$ mass in the invalid part of space, where $P$ has none. Thus, if one can guarantee that $\hat{q}$ has $d_{TV}(\hat{q}, P) \leq \epsilon_2$, the validity requirement is met. Recalling the Pinsker inequality $d_{TV}(q, q') \leq O(\sqrt{d_{KL}(q, q')})$ relating total variational distance and KL-divergence, it follows that obtaining a model $\hat{q}$ which is at most $\epsilon_2^2$ sub-optimal in log-loss yields a model meeting the validity requirement.

While this illustrates useful intuition for the setting, it glosses over two main issues. Firstly, empirical estimates of the log-loss do not admit concentration guarantees – one can construct simple examples where $\mathbb{E}_{X \sim P}[\log(1/f_q(X))]$ is unbounded above, but with high probability, the empirical estimate $L_S(q) = \sum_{x \in S} \log(1/f_q(x))/|S|$ is approximately that of $P$ [8]. Thus, selecting low-empirical loss models can never yield loss guarantees. Secondly, this application of Pinsker's inequality demands $\epsilon_2^2$ loss sub-optimality, suggesting that ensuring validity via the selection of a good model under the log-loss is even harder than guaranteeing a small loss.

---

**Algorithm 1** Modifying ERM to Yield Log-Loss Guarantees

---
1: **procedure** FINITE_LOG_LOSS(Distribution Class $\mathcal{Q}, S, \epsilon_\wedge = \min(\epsilon_1, \epsilon_2)$)
2: $\quad \hat{q}_{\text{ERM}} \leftarrow \arg\min_{q \in \mathcal{Q}} \sum_{x_i \in S} \log\left(1/f_q(x_i)\right)$
3: $\quad$ **return** $\hat{q} = (1 - \epsilon_\wedge/8) \cdot \hat{q}_{\text{ERM}} + \epsilon_\wedge/8 \cdot u$ $\qquad\qquad$ ▷ Mix ERM, uniform distribution
4: **end procedure**

---

We would hope that in the case that zero-query learning is possible, that guaranteeing validity arises somewhat coincidently with convergence to $P$, meaning that the sample complexity is not much worse given a validity requirement than without one. Thus, the path towards satisfaction of the learning objectives requires subtle handling, and compels particular attention to the sample complexity dependence on the validity parameter $\epsilon_2$.

## 4.2 Analysis of Empirical Risk Minimization

As indicated above, it is not possible to guarantee that empirical risk minimization (ERM) outputs a model with small log-loss. It is, however possible to guarantee that it outputs a model which closely resembles $P$ and inherits validity guarantees with a small number of samples.

In particular, it's possible to show that given sufficient samples, ERM yields a model with small total variation to $P$ when $P \in \mathcal{Q}$. This is due to the following folklore theorem [8], which we prove under assumption of density existence in the Appendix.

**Lemma 4.** *Fix $0 < \epsilon, \delta < 1$ arbitrarily, and let $P, q \in \mathcal{P}$ be distributions with densities with respect to $\lambda$. Then if $d_{TV}(q, P) \geq \epsilon$, and $S \sim P^n$ for $n \geq \Omega(\log(1/\delta)/\epsilon^2)$, it holds with probability $\geq 1 - \delta$ that*

$$L_S(P) < L_S(q).$$

Thus, at the statistical cost of estimating a coin bias, any distribution $q$ with total variation $\geq \epsilon$ from the data distribution will reveal itself to be empirically inferior when the log-loss is used. This can be easily leveraged to generate guarantees for ERM over $\mathcal{Q}$ in terms of total variation.

It is tempting to think that this is the entire story when it comes to guaranteeing validity. After all, we argued above that small total variation from $P$ is sufficient for $\epsilon_2$ invalidity. That said, simply looking at total variation ignores a particular structural feature of distributions $q$ with $I(q) > \epsilon_2$ – in particular, such distributions have mass in parts of space in which $P$ does not.

This observation can be used to construct tight lower bounds on the total variational distance between product measures arising from $P$ and $q$ with $I(q) > \epsilon_2$. This leads to the following result, which states that ERM yields a faithful representation of the data generating distribution that is at most $\epsilon_2$ invalid given a number of samples with a modest dependence on the validity parameter $\epsilon_2$.

**Lemma 5.** *Fix $0 < \delta, \epsilon_1, \epsilon_2 < 1$ arbitrarily, and suppose $P \in \mathcal{Q}$. If $P$ is fully-valid under $v$, and $S \sim P^n$ for $n \geq \Omega\left(\frac{\log(|\mathcal{Q}|) + \log(1/\delta)}{\min(\epsilon_1^2, \epsilon_2)}\right)$, then with probability $\geq 1 - \delta$ over $S \sim P^n$, the ERM solution*

$$\hat{q} = \arg\min_{q \in \mathcal{Q}} \sum_{x_i \in S} \log(1/q(x_i)),$$

*satisfies both*

$$d_{TV}(\hat{q}, P) \leq \epsilon_1 \ \ and \ \ I(\hat{q}) \leq \epsilon_2.$$

Note that this guarantee is not redundant – having $d_{TV}(\hat{q}, P) \leq \epsilon_1$ does not imply $I(q) \leq \epsilon_2$ when $\epsilon_2 < \epsilon_1$.

## 4.3 Attaining Log-Loss Guarantees

This result can be interpreted as a vote of confidence for the naive training of generative models under the log-loss. Nevertheless, from a learning-theoretic perspective, there is a question whether or not it is possible to guarantee low log-loss while maintaining validity with zero validity queries.

While ERM cannot possibly furnish log-loss guarantees, it turns out that it is possible to modify the output of ERM to generate log-loss guarantees at the cost of an extra polylogarithmic factor in the sample complexity, at least when the densities $f_q$ are bounded above and below in their support.[1]

The idea, formalized in Algorithm 1, is simply to mix the output of ERM with the uniform distribution. Giving the uniform distribution a mixture component on the order of $\min(\epsilon_1, \epsilon_2)$ can be shown to ensure that the validity guarantees of the ERM output are preserved, while also giving the outputted distribution support across the entire space. This leads to the following theorem.

**Theorem 1.** *Fix $0 < \delta, \epsilon_1, \epsilon_2 < 1$ arbitrarily, and suppose $P \in \mathcal{Q}$ and that $P$ is fully-valid under $v$. If it holds that for each $q \in \mathcal{Q}$ that $\alpha \leq f_q(x) \leq \beta$ for all $x \in \mathrm{supp}(q)$, then there is an*

$$N \leq \tilde{O}\left(\frac{\log^2\left(1/\min(\epsilon_1, \epsilon_2, \alpha)\right) \cdot \left(\log(|\mathcal{Q}|) + \log(1/\delta)\right)}{\min(\epsilon_1^2, \epsilon_2)}\right),$$

*such that for all $n \geq N$, with probability $\geq 1 - \delta$, the output $\hat{q}$ of Algorithm 1 satisfies*

$$L_P(\hat{q}) \leq L_P(q^*) + \epsilon_1 \quad and \quad I(\hat{q}) \leq \epsilon_2.$$

Here the $\tilde{O}$ notation hides a polylogarithmic dependence on $1/\beta$, which is insignificant in most regimes, and treats the density of the uniform distribution over $\mathcal{X}$ as a constant, which would otherwise also enter polylogarithmically.

Theorem 1 shows that guarantees with respect to the unbounded log-loss are attainable improperly, i.e. when the learner can choose $q \notin \mathcal{Q}$. It's an interesting question whether the logarithmic dependence on $\min(\epsilon_1, \epsilon_2)$ can be removed with a more subtle strategy.

## 4.4 Discussion of Optimality

One might suspect that achieving a smaller dependence than $1/\epsilon_2$ on the validity parameter should be impossible. We confirm this is true at least for proper learners, showing that the analysis of ERM in Lemma 5 is tight in its dependence on $\epsilon_2$. This lemma is used to generate the validity guarantee in Theorem 1.

**Theorem 2.** *For all $\epsilon_2 < 1/4$ and for all proper learners $L : S \to \mathcal{P}$, if the sample $S \sim P^n$ is of size $n \leq 1/8\epsilon_2$, then there exists a triple $(P, \mathcal{Q}, v)$ with $P \in \mathcal{Q}$ and $P$ fully-valid, on which $L(S)$ has invalidity $I(L(S)) > \epsilon_2$ with probability $\geq 1/4$.*

The intuition here is that while any invalid $q \in \mathcal{Q}$ has at least $\epsilon_2$ total variation from $P$, in the worst case, the total variation between $q$ and $P$ is upper bounded by $O(\epsilon_2)$ as well. This makes distinguishing between $P$ and some $\epsilon_2$ invalid distribution hard enough to generate such a lower bound.

The sample requirement of $1/\epsilon_1^2$, both in our guarantees and in previous work, is a standard offshoot of loss estimation, irrespective of the search of a valid model. In general, one cannot expect improvements to this end – this is the standard dependence one finds for estimating the means bounded random variables. This suggests that the "realizable complexity" for this setting is $1/\min(\epsilon_1^2, \epsilon_2)$ – while non-zero losses should not be generally estimable using "realizable" techniques, guaranteeing small invalidity can when $P \in \mathcal{Q}$.

## 5 Utilizing Estimates of the Validity Function in Training

In the general formulation of the problem, the learner is given an arbitrary bounded, decreasing loss, a model class $\mathcal{Q}$ which is mismatched with $P$, and has no a priori information about the validity function $v$. In such a setting, it is clear that validity queries are necessary.

In this section, we consider a setting where it is known to the learner that $v$ can be found in a hypothesis class $\mathcal{V}$ of bounded complexity. Under such an assumption, we would hope to be able to lower the number of validity queries beyond the bounds of [3].

---
**Algorithm 2** Post-Hoc Restriction of ERM to an Estimate of Valid Outputs
---
1: **procedure** VALID_RESTRICTION(Distribution Class $\mathcal{Q}$, Validity Class $\mathcal{V}$, $\epsilon_1$, $\epsilon_2$, $\delta$, $\gamma$)

2: $\quad S \leftarrow \Omega\left(\frac{M^2(\log(|Q|)+\log(1/\delta))}{\epsilon_1^2}\right)$ i.i.d. samples $\sim P$

3: $\quad \hat{q}_{\text{ERM}} \leftarrow \arg\min_{q \in \mathcal{Q}} \sum_{x \in S} l(q(x))$

4: $\quad S_P \leftarrow \Omega\left(\frac{M(D\log(M/\epsilon_1)+\log(1/\delta))}{\epsilon_1}\right)$ i.i.d. samples $\sim P$,

$\qquad S_{\hat{q}_{\text{ERM}}} \leftarrow \Omega\left(\frac{D\log(1/\gamma\epsilon_2)+\log(1/\delta)}{\gamma\epsilon_2}\right)$ i.i.d. samples $\sim \hat{q}_{\text{ERM}}$

5: $\quad \hat{v} \leftarrow \arg\min_{h \in \mathcal{V}} \sum_{x \in S_P \cup S_{\hat{q}_{\text{ERM}}}} \mathbb{1}[h(x) \neq v(x)]$ $\qquad\qquad \triangleright$ Label $S_{\hat{q}_{\text{ERM}}}$ via queries to $v$

6: $\quad$ **return** $f_{\hat{q}} \propto f_{\hat{q}_{\text{ERM}}}(x) \cdot \mathbb{1}[\hat{v}(x)=1]$ **if** $\hat{q}_{\text{ERM}}(\{\hat{v}(x)=1\}) > 0$ **else** $f_{\hat{q}} = f_{\hat{q}_{\text{ERM}}}$

7: **end procedure**
---

## 5.1 Algorithm

A natural algorithm in this setting is to "correct" the invalidity of the empirical risk minimizer – to restrict the empirical risk minimizer to parts of space which are valid with respect to an estimate of the validity $\hat{v}$. This is the precisely the idea formalized in Algorithm 2.

To generate guarantees for such a strategy, one must determine the distribution with respect to which the estimate $\hat{v}$ should be accurate. In our case, we generate accuracy guarantees over both $P$ and the ERM model $\hat{q}_{\text{ERM}}$ by selecting an estimate $\hat{v}$ that has 0 empirical error over both distributions. The source of the query complexity of the algorithm comes from the fact that samples arising from $\hat{q}_{\text{ERM}}$ must be labeled by oracle calls to $v$. Noting that samples from $P$ can be automatically labeled as valid by the full-validity of $P$ saves a constant factor over naively labeling all examples acquired in the second half of the algorithm.

Accuracy under samples from $P$ allows one to control the loss of $\hat{q}$ by invoking the boundedness of the loss in the disagreement region of $\hat{v}$ and $v$, and guarantees with respect to $\hat{q}_{\text{ERM}}$ allow us to bound the invalidity of the restriction. Because $P$ is fully-valid, loss contributions from the agreement region of $v$ and $\hat{v}$ correspond to parts of space where $\hat{v}(x) = 1$ – as the loss is non-increasing, placing more mass in such parts of space can never increase the loss contribution attributable to integrating over this region.

Algorithm 2 also requires a parameter $\gamma > 0$. This parameter should be a validity lower bound on the models $q \in \mathcal{Q}$, providing a safeguard on the possibility of an "invalidity blowup" when restricting the ERM output to a certain region of space – one must normalize the restriction to output a probability distribution, which in this case means increasing mass in parts of space that are estimated to be valid. An a priori lower bound on the validity allows for precise enough estimation of $\hat{v}$ that increasing the mass in such regions is unlikely to lead to appreciable invalidity in the final model.

It's possible that the restriction of the ERM estimated valid region is undefined – this happens if and only if the estimated valid region has zero mass under the ERM. Given validity lower bounds for models $q \in \mathcal{Q}$, this is a low probability event which can occur only when estimation of the validity function is very poor relative to the query complexity. As one might imagine, the handling of this case is immaterial for PAC-guarantees. We choose to arbitrarily define behavior in this case by outputting the ERM model.

## 5.2 Guarantees

The restricted output $\hat{q}$ of Algorithm 2 admits the following guarantee over loss sub-optimality and invalidity.

---

[1]This does not yield uniform convergence over $\mathcal{Q}$ given that the support of $q \in \mathcal{Q}$ need not align with $P$

**Theorem 3.** *Suppose $v \in \mathcal{V}$ with VC-dimension $VC(\mathcal{V}) \leq D$, and that for each $q \in \mathcal{Q}$, the validity $V(q) \geq \gamma > 0$. For all $0 < \epsilon_1, \epsilon_2, \delta \leq 1$ and for all choices of non-increasing loss functions $l : \mathbb{R}^{\geq 0} \to [0, M]$, Algorithm 2 requires a number of samples*

$$\leq O\left( \frac{M^2 \left(\log(|\mathcal{Q}|) + \log(1/\delta)\right)}{\epsilon_1^2} + \frac{M \left(D \log(M/\epsilon_1) + \log(1/\delta)\right)}{\epsilon_1} \right),$$

*and a number of validity queries*

$$\leq O\left( \frac{D \log(1/\gamma \epsilon_2) + \log(1/\delta)}{\gamma \epsilon_2} \right),$$

*to ensure that with probability $\geq 1 - \delta$, its output enjoys*

$$L_P(\hat{q}; l) \leq L_P(q^*; l) + \epsilon_1 \ \text{ and } \ I(\hat{q}) \leq \epsilon_2.$$

Thus, in regimes where e.g. $\gamma \geq \Omega(\epsilon_1)$, $D = \Theta(\log(|\mathcal{Q}|))$, this guarantee represents a reduced number of queries under the $\tilde{O}(M^2 \log(|\mathcal{Q}|)/\epsilon_1^2 \epsilon_2)$ bound of [3]. It also implies a "decoupling" of the query complexity from $\epsilon_1$.

We note that the sample requirement from $P$ is increased in certain regimes over the $\tilde{O}(\log(|Q|)/\epsilon_1^2)$ requirement of [3]. This is, however, not a concern in most settings where validity queries are expensive. If samples from a fully-valid $P$ are readily obtainable, the setting is analogous to that of active learning, where focus is directed to the number of labels requested in training.

Even if $P$ must be constructed by "accepting" valid samples from some unfiltered $P'$, a comparison between the query complexity of Theorem 3 and the query bound of [3] is often still representative of the relative data costs of the schemes. Supposing $P'$ produces valid samples with constant probability, the total number of validity queries made by each scheme is proportional to the scheme's sample requirements from $P$, plus the number of validity queries used in its execution. Essentially, to yield validity query speedups, our scheme requires a VC bound on $\mathcal{V}$ which does not dwarf $\log(|\mathcal{Q}|)$. Thus, in most cases of interest, the querying the validity of $\tilde{O}(MD/\epsilon_1)$ extra samples is asymptotically inconsequential relative to $O(M^2 \log(|Q|)/\epsilon_1^2)$, and the $\tilde{O}(D/\gamma \epsilon_2)$ query budget required to execute the algorithm given access to a fully-valid $P$.

## 5.3 Better Query Complexity Bounds

### 5.3.1 Exploiting the Power of Active Learning

Theorem 3 presents a somewhat pessimistic view of the potential of such a "post-filtering" scheme.

Firstly, it ignores the potential of active learners to improve query complexities over passive sampling. Query complexities in active learning of classifiers are often expressed in terms of the "disagreement coefficient" [26], often denoted via $\theta$. In the realizable setting, query complexities of active learning look like $\tilde{O}(D\theta \log(1/\epsilon))$ [27]. Definitionally, it can be shown that $\theta \leq 1/\epsilon$. Thus, proving the gains of active learning algorithms usually relies on bounding the disagreement coefficient non-trivially, i.e. showing $\theta < o(1/\epsilon)$, or ideally, $\theta \leq O(1)$.

While this is challenging, as $\theta$ is both a class and distribution-dependent quantity, there is a literature that addresses this potential in various settings – see the references in [25]. In principle, one could use such an analysis to show that the query complexity of an active learning modification of Algorithm 2 is on a lower order than the guarantee of Theorem 3 when conditions are favorable. To this end, it may be useful to note that a modification of Algorithm 2 wherein $\hat{v}$ is selected as the ERM on a dataset generated by a mixture of $P$ and $\hat{q}_{\text{ERM}}$ admits guarantees as well.

### 5.3.2 Only Low-Loss Models Need Appreciable Validity

Another source of potential looseness in the statement of Theorem 3 is that it phrases the query complexity in terms of the worst-case validity over models $q \in \mathcal{Q}$. This is unnecessary – with high probability, in the first step of the algorithm, one selects a model $q \in \mathcal{Q}$ with $O(\epsilon_1)$ true loss. Thus, what really matters for such a strategy is that models that have relatively small loss $l$ do not have invalidity nearing 1.

This is a realistic scenario in the case that the loss function $l$ – despite possibly not being proper – prioritizes models which in some sense resemble the data generating distribution $P$. Indeed, it's somewhat difficult to envision a situation where a loss would be chosen that prioritizes models with no relation to the data generating distribution. To this end, we give the following corollary to Theorem 3.

**Corollary 1.** *Under the conditions of Theorem 3, if in addition it holds that all models $q \in \mathcal{Q}$ with loss sub-optimality $\epsilon_1$ have validity greater than some constant, then the query complexity of Algorithm 2 can be improved to*

$$\leq O\left(\frac{D\log(1/\epsilon_2) + \log(1/\delta)}{\epsilon_2}\right).$$

### 5.3.3 Removing the Positive Validity Requirement

Using an idea found in the algorithm of [3], one can show that if a learner has access to single distribution $\mathcal{D}$ with a density and at least some non-zero constant validity, and the densities $f_q$ are bounded above, that Algorithm 2 can be modified so as to drop the requirement of positive validity over models when learning under the capped log-loss.

By mixing the $\hat{q}_{\mathrm{ERM}}$ with $\mathcal{D}$, giving mixture component $O(\epsilon_1)$ to $\mathcal{D}$, one can generate similar guarantees as those of Theorem 3. The modification can be found in the Appendix as Algorithm 3, and enjoys the following guarantee.

**Theorem 4.** *Suppose $v \in \mathcal{V}$ where $VC(\mathcal{V}) \leq D$, and that for each $q \in \mathcal{Q}$, we have $f_q(x) \leq \beta$. Suppose further that there is some known $\mathcal{D} \in \mathcal{P}$ with density $f_{\mathcal{D}}$ which has $V(\mathcal{D}) \geq c > 0$ for some constant $c$. Then for all choices of $0 < \epsilon_1, \epsilon_2, \delta < 1/2$ and $M > 0$, Algorithm 3 requires a number of samples*

$$\leq \tilde{O}\left(\frac{M^2\left(\log(|\mathcal{Q}|) + \log(1/\delta)\right)}{\epsilon_1^2} + \frac{M\left(D\log(M/\epsilon_1) + \log(1/\delta)\right)}{\epsilon_1}\right),$$

*and a number of validity queries*

$$\leq O\left(\frac{D\log(1/\epsilon_1\epsilon_2) + \log(1/\delta)}{\epsilon_1\epsilon_2}\right),$$

*to ensure that with probability $\geq 1 - \delta$, its output enjoys*

$$\mathbb{E}_{X \sim P}\left[\min\left(M, \ \log(1/f_{\hat{q}}(X))\right)\right] \leq \mathbb{E}_{X \sim P}\left[\min\left(M, \ \log(1/f_{q^*}(X))\right)\right] + \epsilon_1 \ \text{ and } \ I(\hat{q}) \leq \epsilon_2.$$

Here, the $\tilde{O}$ notation again hides factors polylogarithmic in $1/\beta$.

Note that the $\tilde{O}$ now appears in the sample complexity. This simply reflects the fact that the $M$-capped log-loss can range between gap $M$ and $\log(1/\beta)$ when working with densities bounded above by $\beta \geq 1$. In the case that densities can be bounded above by 1, as in the discrete setting of [3], this dependence disappears.

## 6 Conclusion

This work is intended as a first-look into settings closer to the common-case, where ensuring validity may be relatively cheap.

A more thorough investigation of the log-loss, as well as capped variants, seems a very relevant line of further inquiry, given the widespread use of this family in practice and its useful information-theoretic properties. A natural extension to the first part of this work would be to consider learning in the agnostic case $P \notin \mathcal{Q}$ under the log-loss, where one would hope to be able to exploit these properties and the validity of training samples to keep the number of validity queries low.

In general, characterizing the sample and query demands of validity-constrained distribution learning is challenging, given that proving lower bounds in general requires arguing against learners with two tools at their disposal – sampling and actively querying validity. Work in this direction will likely require some creative constructions.

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

# 7 Appendix

## 7.1 Probability Distributions and Measure Theoretic Formalism

We work over Euclidean space $\mathbb{R}^d$ and let $\mathcal{X}$ be a Lebesgue measurable set with Lebesgue measure $\lambda(\mathcal{X}) < \infty$. By $\lambda$ and Lebesgue measurable set, we refer to the measure and $\sigma$-algebra $\mathcal{F}$ arising from the usual construction of Lebesgue measure on $\mathbb{R}^d$. By "distribution", we mean a probability measure on the measurable space $(\mathcal{X}, \mathcal{F}_{\mathcal{X}})$, where $\mathcal{F}_{\mathcal{X}} = \{E \cap \mathcal{X} : E \in \mathcal{F}\}$. Let $u$ the uniform distribution be the measure given by $u(E) = \lambda(E)/\lambda(\mathcal{X})$ for $E \in \mathcal{F}_{\mathcal{X}}$. Let $\mathcal{P}$ denote the set of all probability measures on $(\mathcal{X}, \mathcal{F}_{\mathcal{X}})$.

We assume that $P \in \mathcal{P}$ and $q \in \mathcal{Q} \subset \mathcal{P}$ have densities $f_P$ and $f_q$ with respect to the reference measure $\lambda$. At times, it will be useful to assume that densities are bounded away from zero in certain parts of space. By saying densities are bounded in their support by $\beta > \alpha > 0$, we mean that for all $x \in supp(q) := \text{cl}\{x : f_q(x) > 0\}$, we have $\alpha \leq f_q(x) \leq \beta$, where the closure is defined through open balls in the Euclidean metric. Note that in this setting, we have $q(supp(q)) = 1$, as $q(supp(q)^c) = \int \mathbb{1}[x \in supp(q)^c] f_q(x) d\lambda(x) = \int 0 d\lambda = 0$.

Denote via $q^n = q \otimes \cdots \otimes q$ the product measure over the measurable space $(\mathcal{X}^{\otimes n}, \mathcal{F}_{\mathcal{X}}^{\otimes n})$. Such a measure corresponds to the process of taking $n$ i.i.d. samples $\sim q$. Denote the density of $q^n$ with respect to $\lambda^n$ via $f_q^n$.

We define $\log(1/0) = \infty$. Following the conventions in [28], we say that $\mathbb{1}[x \in E] \cdot g(x) = 0$ if $g(x) < \infty$ for $x \in E$ and $g(x) = \infty$ for some $x \in E^c$. This allows us to integrate over the finite part of functions and get a finite result.

To facilitate digestibility, we refrain from measure theoretic notation as much as possible. It is at times useful, particularly in dealing with total variation. We assume throughout that all functions we encounter in the Appendix – including the fixed validity function $v$ and functions in the validity class $\mathcal{V}$ – are $(\mathcal{X}, \mathcal{F}_{\mathcal{X}})$-measurable.

## 7.2 Estimates of Validity, Invalidity

We fix the validity function $v$ as an arbitrary function $v : \mathcal{X} \to \{0, 1\}$ measurable with respect to each distribution $q$ arising in the Appendix. As discussed above, for a given model $q \in \mathcal{P}$, the "validity" of $q$ is the quantity $V(q) = \Pr_{X \sim q}(v(X) = 1)$, and the "invalidity" $I(q) = 1 - V(q)$. We will at times be interested in estimating the validity of a model $q$ using samples from $q$ along with validity queries. Given an i.i.d. sample $\{X_i\}_{i=1}^n$ from $q$, we let $\hat{V}(q) = \sum_{i=1}^n v(X_i)/n$ be the natural estimate of the validity of $q$.

At times, we will be interested in the validity of a model under an estimate of the underlying validity function. To this end, given a model $q \in \mathcal{P}$ and a function $g : \mathcal{X} \to \{0, 1\}$, we let $V_g(q) = \Pr_{X \sim q}(g(X) = 1)$. Given a sample $\{X_i\}_{i=1}^n$, let $\hat{V}_g(q) = \sum_{i=1}^n g(X_i)/n$. Note that in the language of this notation, we have $V(q) = V_v(q)$ and $\hat{V}(q) = \hat{V}_v(q)$. We extend this notation in the natural way to invalidity quantities.

## 7.3 Analysis of Empirical Risk Minimization, Improper Algorithm in Realizable Setting

To begin our analysis of the realizable setting, we first observe that models with appreciable invalidity look very different from a fully-valid data generating distribution – because they must have mass in parts of space where $P$ does not, they are separated in total variation from $P$ by a margin. We formalize this idea via the following.

**Lemma 1.** *Fix $0 < \epsilon < 1$ arbitrarily. For any validity function $v$, if $q \in \mathcal{P}$ has $I(q) > \epsilon$, and $P \in \mathcal{P}$ has $I(P) = 0$, then*

$$d_{TV}(P, q) > \epsilon.$$

*Proof.* Fix the validity function arbitrarily. Consider the event $E_{\neg v} = \{x \in \mathcal{X} : v(x) = 0\}$. Then we have $d_{TV}(P, q) = \sup_{E \subseteq \mathcal{X}} |P(E) - q(E)| \geq q(E_{\neg v}) - P(E_{\neg v}) > \epsilon$, where we have used that $I(P) = 0$ implies $P(E_{\neg v}) = 0$. $\qquad\square$

We next extend this observation to the associated product measures, which are the main target of analysis under i.i.d. sampling from $P$. The idea is to lower bound the total variation between product measures by the difference in probabilities on the event that at least one example from a sample of size $n$ is invalid. Of course, for each sample, this happens with probability 0 under $P$ and with probability at least $\epsilon$ under any model $q$ with at least $\epsilon$ invalidity. Thus, identically to how one shows realizable rates for classification tasks, we attain a large total variation gap between $P$ and any $q$ with appreciable invalidity – mistakes in classification are thus analogous to the generation of an "invalid" samples in our setting.

**Lemma 2.** *Fix $0 < \epsilon < 1$ and $n \in \mathbb{N} \setminus \{0\}$ arbitrarily. For any validity function $v$, if $q \in \mathcal{P}$ has $I(q) > \epsilon$, and $P \in \mathcal{P}$ has $I(P) = 0$, then*

$$d_{TV}(P^n, q^n) > 1 - e^{-n\epsilon}.$$

*Proof.* Fix the validity function arbitrarily. Consider lower bounding the total variation between the product measures via the magnitude of the difference of their measures on the event

$$E_{\geq 1} = \left\{ (x_1, \ldots, x_n) \in \mathcal{X}^n : \exists i \text{ s.t. } v(x_i) = 0 \right\}.$$

Because $P$ has perfect validity, any given draw from $P$ has probability 0 of being invalid. Thus, $P^n(E_{\geq 1}) = 0$. On the other hand, the invalidity of $q$ states that for any $1 \leq i \leq n$, we have $q(\{x \in \mathcal{X} : v(x) = 0\}) > \epsilon$. Let $E_v = \{x \in \mathcal{X} : v(x) = 1\}$, and note that $q(E_v) < 1 - \epsilon$. Then we have

$$
\begin{aligned}
q^n(E_{\geq 1}) &= 1 - q(E_v)^n \\
&> 1 - (1 - \epsilon)^n \\
&\geq 1 - e^{-n\epsilon},
\end{aligned}
$$

where the final inequality follows from the fact that $(1 + x/n)^n \leq e^x$ for $x \leq n$. $\qquad \square$

We can then borrow from classical analysis of hypothesis testing given by the Neyman-Pearson lemma to leverage this gap in total variation between product measures into a bound on the probability that after $n$ samples, a model with appreciable invalidity has a smaller loss than $P$.

**Lemma 3.** *Fix $0 < \epsilon < 1$ and $n \in \mathbb{N} \setminus \{0\}$ arbitrarily. For any validity function $v$, if $q \in \mathcal{P}$ has $I(q) > \epsilon$, $P \in \mathcal{P}$ has $I(P) = 0$, and $q$ and $P$ have densities with respect to the reference measure $\lambda$, then*

$$\Pr_{S \sim P^n} \left( q^n(S) \geq P^n(S) \right) \leq e^{-n\epsilon}.$$

*Proof.* The proof follows that of the Neyman-Pearson lemma's claim that the Likelihood Ratio Test achieves the lower bound on the sum of Type I and Type II errors [18], combined with Lemma 2. We give the full argument for completeness.

Fix the validity function arbitrarily, and note the following string of relations:

$$
\begin{aligned}
\Pr_{S \sim P^n} \left( q^n(S) \geq P^n(S) \right) &= \int \mathbb{1}[f_q^n(x) \geq f_P^n(x)] f_P^n(x) d\lambda^n(x) \\
&\leq \int \min \left( f_q^n(x), f_P^n(x) \right) d\lambda^n(x) \\
&= 1 - d_{TV}(P^n, q^n) \\
&\leq e^{-n\epsilon},
\end{aligned}
$$

where the switch to total variation in the second to last line is the result of a classic characterization of total variation given by "Scheffé's Theorem", and the final line comes from Lemma 2. $\qquad \square$

To generate loss guarantees, we need to be able to reason about models which do not have appreciable invalidity. The next lemma is an analogue of the previous one – it is identical up to the replacement of the condition that $q$ have appreciable invalidity with a weaker one that the total variation between a model $q$ and the data distribution $P$ is appreciable. When $q$ does not have appreciable invalidity, we can no longer rely on the structure of the supports of $q$ and $P$ to generate large margins for the total variation separation of product measures, and have to fall back on more general estimates for total variation between product measures.

**Lemma 4.** *Fix $0 < \epsilon, \delta < 1$ arbitrarily, and let $P, q \in \mathcal{P}$ be distributions with densities with respect to $\lambda$. Then if $d_{TV}(q, P) \geq \epsilon$, and $S \sim P^n$ for $n \geq \Omega(\log(1/\delta)/\epsilon^2)$, it holds with probability $\geq 1 - \delta$ that*

$$L_S(P) < L_S(q).$$

*Proof.* When $q$ and $P$ both possess densities, we can related the the probability that the likelihood of $q$ is at least that of $P$ to their total variation, as in the Neyman-Pearson Lemma.

$$\Pr_{S \sim P^n}\left(P^n(S) \leq q^n(S)\right) = \int \mathbb{1}[f_P^n(x) \leq f_q^n(x)]f_P^n(x)d\lambda^n(x)$$

$$\leq \int \min\left(f_P^n(x), f_q^n(x)\right) d\lambda^n(x)$$

$$= 1 - d_{TV}(P^n, q^n)$$

$$\leq e^{-n d_{TV}(p,q)^2/2}$$

$$\leq e^{-n\epsilon^2/2}.$$

Here, the second to last inequality is the consequence of powerful result of [19] (and later [29]), namely that for any two collections of probability measures $\{q_i\}_{i=1}^n$ and $\{p_i\}_{i=1}^n$ over measurable spaces $\{(\mathcal{X}_i, \mathcal{F}_i)\}_{i=1}^n$, the product measures over the respective collections $q^n$ and $p^n$ satisfy

$$1 - \exp\left(-\frac{1}{2}\sum_{i=1}^n d_{TV}(q_i, p_i)^2\right) \leq d_{TV}(q^n, p^n).$$

The final inequality follows from assumed gap in total variation between $P$ and $q$. $\square$

The previous two results concern the testing of individual models against $P$. In the standard way, we now leverage the finite cardinality to argue via a union bound that given enough samples from $P$, it's unlikely that the ERM model has a large total variation distance from $P$.

**Lemma 5.** *Fix $0 < \delta, \epsilon_1, \epsilon_2 < 1$ arbitrarily, and suppose $P \in \mathcal{Q}$. If $P$ is fully-valid under $v$, and $S \sim P^n$ for $n \geq \Omega\left(\frac{\log(|\mathcal{Q}|) + \log(1/\delta)}{\min(\epsilon_1^2, \epsilon_2)}\right)$, then with probability $\geq 1 - \delta$ over $S \sim P^n$, the ERM solution*

$$\hat{q} = \arg\min_{q \in \mathcal{Q}} \sum_{x_i \in S} \log(1/q(x_i)),$$

*satisfies both*

$$d_{TV}(\hat{q}, P) \leq \epsilon_1 \quad and \quad I(\hat{q}) \leq \epsilon_2.$$

*Proof.* Let $\mathcal{Q}_{d_{TV} > \epsilon_1} = \{q \in \mathcal{Q} : d_{TV}(q, P) > \epsilon_1\}$. By Lemma 4 and a union bound, it holds that

$$\Pr_{S \sim P^n}\left(d_{TV}(\hat{q}, P) > \epsilon_1\right) \leq \Pr_{S \sim P^n}\left(\exists q \in \mathcal{Q}_{d_{TV} > \epsilon_1} \; s.t. \; L_S(q) \leq L_S(P)\right)$$

$$\leq |\mathcal{Q}_{d_{TV} > \epsilon_1}| \cdot \max_{q \in \mathcal{Q}_{d_{TV} > \epsilon_1}} \Pr_{S \sim P^n}\left(L_S(q) \leq L_S(P)\right)$$

$$\leq |\mathcal{Q}|e^{-n\epsilon_1^2/2}.$$

In the same manner, let $v$ be an arbitrary validity function, and let $\mathcal{Q}_{I > \epsilon_2} = \{q \in \mathcal{Q} : I(q) > \epsilon_2\}$. Then we have

$$\Pr_{S \sim P^n}\left( I(\hat{q}) > \epsilon_2 \right) \leq \Pr_{S \sim P^n}\left( \exists q \in \mathcal{Q}_{I > \epsilon_2} \ s.t. \ L_S(q) \leq L_S(P) \right)$$

$$\leq |\mathcal{Q}_{I > \epsilon_2}| \cdot \max_{q \in \mathcal{Q}_{I > \epsilon_2}} \Pr_{S \sim P^n}\left( L_S(q) \leq L_S(P) \right)$$

$$\leq |\mathcal{Q}| \, e^{-n\epsilon_2},$$

where the final inequality holds by Lemma 3. Then finally,

$$\Pr_{S \sim P^n}\left( d_{TV}(\hat{q}, P) > \epsilon_1 \ \vee \ I(\hat{q}) > \epsilon_2 \right) \leq |\mathcal{Q}| e^{-n\epsilon_1^2/2} + |\mathcal{Q}| e^{-n\epsilon_2},$$

and so choosing $n \geq 2(\log(|\mathcal{Q}|) + \log(1/\delta))/\min(\epsilon_1^2, \epsilon_2)$ ensures that the sum of these final terms is $\leq \delta$. $\qquad\square$

Before we can prove Theorem 1, we need one final intermediate result, which we now give. It states that the value of the log-loss at any given $x$ for a mixture distribution constructed by heavily weighting one of two distributions is not much different than the value of the loss for the heavily weighted component. This follows from the fact that the natural log is well-approximated by a linear function near 1. We will use this result to argue that the loss of the output of Algorithm 1 is not significantly different than that of the ERM in the support of the ERM.

**Lemma 4.** *Fix* $0 < \epsilon < 1$. *For any* $q \in \mathcal{P}$ *having a density with respect to* $\lambda$, *the mixture* $M = (1 - \epsilon/2)q + \epsilon u/2$ *has density* $f_M(x) = (1 - \epsilon/2)f_q(x) + \epsilon f_u(x)/2$ *and this density satisfies*

$$\log\left( 1/f_M(x) \right) \leq \epsilon + \log\left( 1/f_q(x) \right),$$

*for all* $x \in \mathcal{X}$.

*Proof.* The existence claim on the densities is immediate given the definition of $u$ as the uniform distribution with respect to the reference measure $\lambda$. To see the inequality, fix $x \in \mathcal{X}$ arbitrarily, and note that $f_M(x) \geq (1 - \epsilon/2)f_q(x)$. Thus, we may write

$$\log\left( 1/f_M(x) \right) \leq \log\left( \frac{1}{1 - \epsilon/2} \right) + \log\left( 1/f_q(x) \right)$$

$$\leq \left( \frac{1}{1 - \epsilon/2} - 1 \right) + \log\left( 1/f_q(x) \right)$$

$$\leq \frac{\epsilon/2}{1 - \epsilon/2} + \log\left( 1/f_q(x) \right)$$

$$\leq \epsilon + \log\left( 1/f_q(x) \right),$$

where second equality comes from the fact that $\log(z) \leq z - 1$, the final follows from the fact that $z/(1 - z) \leq 2z$ for $z \leq 1/2$. $\qquad\square$

To get guarantees for the log-loss, we make heavy use of the previous lemma. Being able to guarantee a small total variational distance from the ERM to $P$ and small invalidity means that the ERM output is already likely to be a faithful representation of $P$ with small invalidity. All that is then needed is to eliminate the possibility that the ERM has a large log-loss because of small mismatches in support with $P$.

To deal with this possibility, we mix the ERM model with the uniform distribution in accordance with Algorithm 1. Because the weight given to the uniform distribution is $O(\min(\epsilon_1^2, \epsilon_2))$, the invalidity is close to that of the ERM. To show that such a move does not increase the loss significantly, we split the contribution to the loss of the outputted model into two that arising from $supp(\hat{q}_{\text{ERM}})$ and it's complement. We first observe that the ERM always has an empirical risk which is a faithful estimator of the integral $\mathbb{E}_{X \sim P}\left[ \mathbb{1}[X \in supp(\hat{q}_{\text{ERM}})] \cdot \log(1/f_{\hat{q}_{\text{ERM}}}(x)) \right]$, allowing us to bound this

integral above in terms of the loss of $P$. The integral of the loss over $supp(\hat{q}_{\text{ERM}})^c$ can be shown to be small using the lower bound on the density of the output model $\hat{q}$ afforded by mixing with the uniform distribution, and the fact that $supp(\hat{q}_{\text{ERM}})^c$ must have a small measure under $P$ when the ERM is close to $P$ in total variation.

**Theorem 1.** *Fix $0 < \delta, \epsilon_1, \epsilon_2 < 1$ arbitrarily, and suppose $P \in \mathcal{Q}$ and that $P$ is fully-valid under $v$. If it holds that for each $q \in \mathcal{Q}$ that $\alpha \leq f_q(x) \leq \beta$ for all $x \in \text{supp}(q)$, then there is an*

$$N \leq \tilde{O}\left(\frac{\log^2\left(1/\min(\epsilon_1, \epsilon_2, \alpha)\right) \cdot \left(\log(|\mathcal{Q}|) + \log(1/\delta)\right)}{\min(\epsilon_1^2, \epsilon_2)}\right),$$

*such that for all $n \geq N$, with probability $\geq 1 - \delta$, the output $\hat{q}$ of Algorithm 1 satisfies*

$$L_P(\hat{q}) \leq L_P(q^*) + \epsilon_1 \quad and \quad I(\hat{q}) \leq \epsilon_2.$$

*Proof.* Fix $v$ arbitrarily, and let $\epsilon_\wedge = \min(\epsilon_1, \epsilon_2)$. To see the claim on the validity, note that by the guarantee of Lemma 5, the assymptotic complexity of Lemma 5 yields the guarantee $I(\hat{q}_{\text{ERM}}) \leq \epsilon_2/2$ with probability $\geq 1 - \delta/3$, in which case

$$\Pr_{X \sim \hat{q}}\left(v(X) = 0\right) = \Pr_{X \sim \hat{q}_{\text{ERM}}}\left(v(X) = 0\right) \cdot \left(1 - \frac{\epsilon_\wedge}{8}\right) + \Pr_{X \sim U}\left(v(X) = 0\right) \cdot \frac{\epsilon_\wedge}{8}$$

$$\leq \frac{\epsilon_2}{2} \cdot \left(1 - \frac{\epsilon_\wedge}{8}\right) + 1 \cdot \frac{\epsilon_\wedge}{8}$$

$$< \epsilon_2.$$

The loss of the outputted model $\hat{q}$ can be decomposed into the contributions from the loss in $\text{supp}(\hat{q}_{\text{ERM}})$ and it's complement:

$$L_P(\hat{q}) = \mathbb{E}_{X \sim P}\left[\mathbb{1}[X \in \text{supp}(\hat{q}_{\text{ERM}})] \cdot \log(1/f_{\hat{q}}(X))\right] + \mathbb{E}_{X \sim P}\left[\mathbb{1}[X \in \text{supp}(\hat{q}_{\text{ERM}})^c] \cdot \log(1/f_{\hat{q}}(X))\right].$$

To bound the first term, for each $q \in \mathcal{Q}$, consider the function

$$B_q(x) = \begin{cases} \log\left(1/f_q(x)\right) & \text{if } x \in supp(q), \\ 0 & \text{else.} \end{cases}$$

These functions are bounded above by $\log(1/\alpha)$ and below by $\log(1/\beta)$ (if $\beta < 1$, simply loosen the density upper bound), and thus for a sample $X \sim P$, define bounded random variables $B_q(X)$. By Hoeffding's inequality and a union bound, it holds that a sample $S$ of size $n \geq \tilde{\Omega}(\log^2(1/\alpha)(\log(|\mathcal{Q}|) + \log(1/\delta))/\epsilon_1^2)$ from $P$ is large enough such that with probability $\geq 1 - \delta/3$, for each $q \in \mathcal{Q}$, it holds that

$$\left| \mathbb{E}_{X \sim P}[B_q(X)] - \frac{1}{n} \sum_{x_i \in S} B_q(x_i) \right| \leq \frac{\epsilon_1}{8}.$$

Note that for each $q \in \mathcal{Q}$ with $L_S(q) < \infty$, it holds that $L_S(q)$ coincides with the empirical estimates of $\mathbb{E}_{X \sim P}[B_q(X)]$, namely

$$\frac{1}{n} \sum_{x_i \in S} B_q(x_i) = \frac{1}{n} \sum_{x_i \in S} \log\left(1/f_q(x_i)\right).$$

Furthermore, this coincidence takes place for $\hat{q}_{\text{ERM}}$ with probability 1, as $P \in \mathcal{Q}$ implies that $L_S(\hat{q}_{\text{ERM}}) \leq L_S(P) < \infty$ with probability 1 – note that $L_S(P)$ is a good estimator for $L_P(P)$ in the sense arising from an application of Hoeffding, as $\log(1/f_P(X))$ is bounded almost surely for $X \sim P$ given that $P$ has a density that is bounded in it's support (as a member of $\mathcal{Q}$). Thus, we may write

$$\mathbb{E}_{X \sim P}\left[\mathbb{1}[X \in \text{supp}(\hat{q}_{\text{ERM}})] \cdot \log(1/f_{\hat{q}}(X))\right] \leq \mathbb{E}_{X \sim P}\left[\mathbb{1}[X \in \text{supp}(\hat{q}_{\text{ERM}})] \cdot \log(1/f_{\hat{q}_{\text{ERM}}}(X))\right] + \frac{\epsilon_1}{4}$$

$$= \mathbb{E}_{X \sim P}\left[B_{\hat{q}_{\text{ERM}}}(X)\right] + \frac{\epsilon_1}{4}$$

$$\leq L_S(\hat{q}_{\text{ERM}}) + \frac{3\epsilon_1}{8}$$

$$\leq L_P(q^*) + \frac{\epsilon_1}{2},$$

where we invoke Lemma 4 in the first step, and in the last step use that $q^* = P$ by the strict properness of the log-loss.

To bound the second term, note that by the argument in given in Lemma 5, when $n \geq \Omega\left(\log^2(1/\epsilon_\wedge) \cdot (\log(|\mathcal{Q}| + \log(1/\delta))/\epsilon_1^2\right)$, it holds with probability $\geq 1 - \delta/3$ that $d_{TV}(\hat{q}_{\text{ERM}}, P) \leq \epsilon_1/8\log(8/\epsilon_\wedge)$. Because $\hat{q}$ has density at least $\epsilon_\wedge/8$ everywhere in $\mathcal{X}$,[2] we have

$$\mathbb{E}_{X \sim P}\left[\mathbb{1}[X \in \text{supp}(\hat{q}_{\text{ERM}})^c] \cdot \log(1/f_{\hat{q}}(X))\right] \leq \log(8/\epsilon_\wedge) \cdot \mathbb{E}_{X \sim P}\left[\mathbb{1}[X \in \text{supp}(\hat{q}_{\text{ERM}})^c]\right]$$
$$\leq \log(8/\epsilon_\wedge) \cdot d_{TV}(P, \hat{q}_{\text{ERM}})$$
$$< \epsilon_1/2,$$

where in the second line we use the fact that $\hat{q}_{\text{ERM}}(supp(\hat{q}_{\text{ERM}})) = 1$ to argue that $d_{TV}(P, \hat{q}_{\text{ERM}}) \geq P(supp(\hat{q}_{\text{ERM}})^c)$. Combining the bounds on the two summands and union bounding the confidence yields the full guarantee.

We note that this argument implies a slight more precise sample complexity bound given by

$$\tilde{O}\left(\max\left(\frac{\log(|\mathcal{Q}|) + \log(1/\delta)}{\epsilon_2}, \frac{\log^2(1/\min(\epsilon_1, \epsilon_2, \alpha))\left(\log(|\mathcal{Q}|) + \log(1/\delta)\right)}{\epsilon_1^2}\right)\right),$$

which affords a minor improvement in certain regimes where $\epsilon_2 < \epsilon_1^2$. $\qquad\square$

### 7.4   A Zero-Query Lower Bound

**Theorem 2.** *For all $\epsilon_2 < 1/4$ and for all proper learners $L : S \to \mathcal{P}$, if the sample $S \sim P^n$ is of size $n \leq 1/8\epsilon_2$, then there exists a triple $(P, \mathcal{Q}, v)$ with $P \in \mathcal{Q}$ and $P$ fully-valid, on which $L(S)$ has invalidity $I(L(S)) > \epsilon_2$ with probability $\geq 1/4$.*

*Proof.* We give an argument inspired by the proof of a lower bound in Theorem 2 of [30].

Let $\mathcal{X} = [0, 1] \subset \mathbb{R}$. Fix the proper learner $L$ and $\epsilon_2 < 1/4$ arbitrarily. Consider a model class defined by $\mathcal{Q} = \{P, \tilde{P}\}$, where $P$ and $\tilde{P}$ have densities with respect to the Lebesgue measure

$$f_P(x) = \mathbb{1}[x \in [0, 1 - 2\epsilon_2]]\frac{1}{1 - 2\epsilon_2}, \quad f_{\tilde{P}}(x) = \mathbb{1}[x \in [2\epsilon_2, 1]]\frac{1}{1 - 2\epsilon_2}.$$

This $\mathcal{Q}$ gives rise to two realizable problem instances of interest. Under the first, $P$ is the data generating distribution and $v(x) = 1$ everywhere except for in $(1 - 2\epsilon_2, 1]$, where $v(x) = 0$. Under the second, $\tilde{P}$ is the data generating distribution and $v(x) = 1$ everywhere except for in $[0, 2\epsilon_2)$, where $v(x) = 0$. Assume by contradiction that for both problem instances, given a sample of size $n \leq 1/8\epsilon_2$, we have that $I(L(S)) \leq \epsilon_2$ with probability $> 3/4$. In both cases, we have that $I(q) = 2\epsilon_2/(1 - 2\epsilon_2) > \epsilon_2$ for the model $q$ which is not the data generating distribution. Thus, $L(S)$ is a model with $I(L(S)) \leq \epsilon_2$ with probability $> 3/4$ over both problem instances if and only if it identifies the data generating distribution with probability $> 3/4$ over both problem instances.

Consider the simple hypothesis tester defined by $T_L(S) = L(S)$, which by the above, outputs the correct data generating distribution given a choice of $P$ or $\tilde{P}$ – and given $n \leq 1/8\epsilon_2$ samples from either $P$ or $\tilde{P}$ – with probability $> 3/4$. Note that the distributions $P, \tilde{P}$ have $d_{TV}(P, \tilde{P}) = 2\epsilon_2/(1 - 2\epsilon_2) \leq 4\epsilon_2$, where we use $\epsilon_2 < 1/4$ and $z/(1 - z) \leq 2z$ for small enough $z$ in the inequality. By a classic upper bound on the total variation between product measures [19], we have that $d_{TV}(P^n, \tilde{P}^n) \leq 4n\epsilon_2$. Then, by Le Cam's method and this upper bound on $d_{TV}(P^n, \tilde{P}^n)$, we have

$$\inf_{T:S \to \{P, \tilde{P}\}} \max_{q \in \{P, \tilde{P}\}} \Pr_{S \sim q^n}(T_L(S) \neq q) \geq \frac{1}{2} - \frac{1}{2} \cdot d_{TV}(P^n, \tilde{P}^n) \geq \frac{1}{2} - 2n\epsilon_2.$$

Thus, if $n \leq 1/8\epsilon_2$, the there is a choice of data generating distribution such that $T_L(S)$ incurs error probability $\geq 1/4$, which is a contradiction. $\qquad\square$

---

[2]When the uniform distribution has a density smaller than 1, there is an extra log factor to account for here. We can WLOG this away by adding the condition that $\lambda(\mathcal{X}) = 1$, e.g. arising from normalized data.

## 7.5 Strategies Arising from Estimation of the Validity Function

When the validity function $v$ is known to lie in a class of bounded complexity, it is learnable, and learned estimates may be utilized in the selection of a low-loss, high-validity model. An interesting feature of the problem setup here is that unlike in most learning settings, the learner can actually *choose* which distributions it would like estimate $v$ with respect to, i.e. decide under which marginal distributions over $\mathcal{X}$ the estimate $\hat{v}$ should small disagreement with $v$.

We begin with a lemma arguing that whenever the ERM has positive probability of outputting an example with $\hat{v}(x) = 1$, and the disagreement of $\hat{v}$ and $v$ under a proposal distribution is small enough, the restriction will indeed yield a low-validity model.

**Lemma 5.** *Fix $0 < \epsilon < 1$ and a distribution $q \in \mathcal{P}$ absolutely continuous with respect to $\lambda$ arbitrarily. Further, fix $\hat{V}(q)$ such that $0 < \hat{V}(q) \leq V(q)$, and suppose that for some $\hat{v} : \mathcal{X} \to \{0, 1\}$ we have*

$$\Pr_{X \sim q} (\hat{v}(X) \neq v(X)) \leq \frac{\hat{V}(q)\epsilon}{2}.$$

*Then whenever there is a distribution $\hat{q}$ corresponding to*

$$f_{\hat{q}}(x) \propto f_q(x) \mathbb{1}[\hat{v}(x) = 1],$$

*it has invalidity $I(\hat{q}) \leq \epsilon$.*

*Proof.* Let $V_{\hat{v}}(q) = \Pr_{X \sim q} (\hat{v}(X) = 1)$ denote the normalizing constant for the restriction to estimated valid region. Note that the restriction corresponds to a probability distribution if and only if $V_{\hat{v}}(q) > 0$, and that in this case

$$f_{\hat{q}}(x) = \frac{f_q(x) \mathbb{1}[\hat{v}(x) = 1]}{V_{\hat{v}}(q)}.$$

It holds further that $\hat{q}$ is absolutely continuous with respect to $\lambda$, and that we can write the following chain of relations:

$$\begin{aligned}
I(\hat{q}) &= \int \mathbb{1}[v(x) = 0] f_{\hat{q}}(x) \, d\lambda(x) \\
&\leq \frac{1}{V_{\hat{v}}(q)} \int \mathbb{1}[\hat{v}(x) \neq v(x)] f_q(x) \, d\lambda(x) \\
&\leq \frac{\hat{V}(q)}{V_{\hat{v}}(q)} \frac{\epsilon}{2} \\
&\leq \frac{V(q)}{V_{\hat{v}}(q)} \frac{\epsilon}{2},
\end{aligned}$$

where the first inequality follows after inputting the definition of $f_{\hat{q}}$, and the final two inequalities come by assumption. It's further possible to show that validity of $q$ can be approximated from above by a constant multiple of $V_{\hat{v}}(q)$, which can be conceptualized as the validity of $q$ if $\hat{v}$ were the true validity function. In particular,

$$\begin{aligned}
V(q) &= \left( V_{\hat{v}}(q) - V_{\hat{v}}(q) \right) + V(q) \\
&= V_{\hat{v}}(q) + \int \left( \mathbb{1}[v(x) = 1] - \mathbb{1}[\hat{v}(x) = 1] \right) f_q(x) \, d\lambda(x) \\
&\leq V_{\hat{v}}(q) + \int \mathbb{1}[v(x) \neq \hat{v}(x)] f_q(x) \, d\lambda(x) \\
&\leq V_{\hat{v}}(q) + \frac{\hat{V}(q)\epsilon}{2} \\
&\leq V_{\hat{v}}(q) + \frac{V(q)\epsilon}{2}.
\end{aligned}$$

This implies that $V(q) \leq V_{\hat{v}}(q)/(1 - \epsilon/2)$, which yields $V(q) \leq 2V_{\hat{v}}(q)$ as $\epsilon < 1$. Utilizing this inequality in the last line of the first string of inequalities gives the guarantee. $\square$

The need for a lower estimate on the validity in the precision of the estimate for $\hat{v}$ can be understood as follows: when the proposal distribution has very small validity, the restriction to the estimation of the valid parts of space under $\hat{v}$ may create huge (or infinite) increases in mass over the proposal distribution – in the language of Lemma 5, $V_{\hat{v}}(q)$ may be very small for a proposal distribution $q$. If this is the case, the estimate $\hat{v}$ must be more precise, as small errors in the estimation of the validity function may lead to invalid parts of space having large mass under $\hat{q}$.

We introduce another lemma before proving Theorem 3. It argues that for a model $\hat{q}$ constructed by accepting samples from some "proposal distribution" $q$ that fall in the valid part of space under $\hat{v}$, the contribution to the loss from the part of space where $\hat{v}$ agrees with $v$ can never exceed the total loss of the proposal distribution $q$. Essentially, we are exploiting the full-validity of $P$ here – in the subregion of the agreement region $\{v(x) = \hat{v}(x)\}$ on which the loss is computed, we have that $\hat{v}(x) = 1$ by the fact that $X \sim P$ is valid. This means that the density of $\hat{q}$ can only be larger than the density of $q$ in this region, which under a non-increasing loss cannot increase the loss over that incurred by $q$.

**Lemma 6.** *Fix $0 < \epsilon, \delta < 1$, a validity function estimate $\hat{v} : \mathcal{X} \to \{0, 1\}$, and $q \in \mathcal{Q}$ arbitrarily. Suppose $l : \mathbb{R}^{\geq 0} \to [0, M]$ is a non-increasing loss function. Then whenever*

$$f_{\hat{q}}(x) \propto f_q(x) \mathbb{1}[\hat{v}(x) = 1]$$

*corresponds to a probability distribution, it enjoys*

$$\mathbb{E}_{X \sim P}\big[l\left(f_{\hat{q}}(X)\right) \cdot \mathbb{1}[\hat{v}(X) = v(X)]\big] \leq L_P(q; l).$$

*Proof.* Let $V_{\hat{v}}(q) = \mathrm{Pr}_{X \sim q}(\hat{v}(x) = 1)$ be the normalizing constant for $\hat{q}$, where we note that $V_{\hat{v}}(q) > 0$ when $f_q(x) \mathbb{1}[\hat{v}(x) = 1]$ corresponds to a probability distribution.

Given that $P$ is fully-valid, we have $\mathrm{Pr}_{X \sim P}(v(X) = 1) = 1$. By the fact that integration is defined up to null sets, it holds that

$$\mathbb{E}_{X \sim P}\left[l\left(f_{\hat{q}}(X)\right) \cdot \mathbb{1}[\hat{v}(X) = v(X)]\right] = \mathbb{E}_{X \sim P}\left[l\left(f_{\hat{q}}(X)\right) \cdot \mathbb{1}[\hat{v}(X) = v(X) \wedge v(X) = 1]\right].$$

Further, we may write

$$\mathbb{E}_{X \sim P}\left[l\left(f_{\hat{q}}(X)\right) \cdot \mathbb{1}[\hat{v}(X) = v(X) \wedge v(X) = 1]\right]$$
$$\leq \mathbb{E}_{X \sim P}\left[l\left(\frac{f_q(X)\mathbb{1}[\hat{v}(X) = 1]}{V_{\hat{v}}(q)}\right) \cdot \mathbb{1}[\hat{v}(X) = 1]\right]$$
$$\leq \mathbb{E}_{X \sim P}\left[l\left(\frac{f_q(X)}{V_{\hat{v}}(q)}\right)\right]$$
$$\leq \mathbb{E}_{X \sim P}\left[l\left(f_q(X)\right)\right]$$
$$= L_P(q; l).$$

Here, the second to last inequality comes from the non-negativity of the loss along with the fact that whenever $\hat{v}(X) = 0$, the integrand is zero – when $\hat{v}(X) = 1$, the loss is just evaluated at the normalized density, and so the integrand introduced in this line is an upper bound for the previous integrand. The final inequality comes from the non-increasingness of the loss function along with the observation that $V_{\hat{v}}(\hat{q}) \leq 1$ – in removing the normalizing constant, we can only make the value at which the loss is evaluated at smaller, which cannot decrease the value of the loss. $\square$

We are now ready to prove the main result of the second half of the paper – the guarantee for Algorithm 2. It combines the previous lemmas, noting further that the number of samples in $S_P$ is sufficient to make the disagreement of $v$ and $\hat{v}$ small enough under $P$ such that the contribution to the loss in that part of space can be controlled by trivially applying the loss upper bound $M$.

**Theorem 3.** *Suppose $v \in \mathcal{V}$ with VC-dimension $VC(\mathcal{V}) \leq D$, and that for each $q \in \mathcal{Q}$, the validity $V(q) \geq \gamma > 0$. For all $0 < \epsilon_1, \epsilon_2, \delta \leq 1$ and for all choices of non-increasing loss functions $l : \mathbb{R}^{\geq 0} \to [0, M]$, Algorithm 2 requires a number of samples*

$$\leq O\left( \frac{M^2 \left(\log(|\mathcal{Q}|) + \log(1/\delta)\right)}{\epsilon_1^2} + \frac{M \left(D \log(M/\epsilon_1) + \log(1/\delta)\right)}{\epsilon_1} \right),$$

*and a number of validity queries*

$$\leq O\left( \frac{D \log(1/\gamma \epsilon_2) + \log(1/\delta)}{\gamma \epsilon_2} \right),$$

*to ensure that with probability $\geq 1 - \delta$, its output enjoys*

$$L_P(\hat{q}; l) \leq L_P(q^*; l) + \epsilon_1 \quad and \quad I(\hat{q}) \leq \epsilon_2.$$

*Proof.* Given that the loss is bounded, Hoeffding's inequality applied to the random variables $l(f_q(X))$ for $X \sim P$, and a union bound, imply that $S$ is large enough that with probability $\geq 1 - \delta/3$, we have that for all $q \in \mathcal{Q}$, the empirical loss estimates $L_S(q; l)$ are at most $\epsilon_1/4$ away from true losses $L_P(q; l)$. For any choice of $\hat{q}_{\mathrm{ERM}}$, because we have $v \in \mathcal{V}$, it must hold that any minimizer $\hat{v}$ is consistent with the labeling under $v$ of both $S_P$ and $S_{\hat{q}_{\mathrm{ERM}}}$. The standard rates of convergence when choosing an arbitrary consistent hypothesis thus imply that the sizes of $S_P$ and $S_{\hat{q}_{\mathrm{ERM}}}$ are large enough to guarantee that, with probability $\geq 1 - 2\delta/3$, we have

$$\mathrm{Pr}_{X \sim P}\left( \hat{v}(X) \neq v(X) \right) \leq \frac{\epsilon_1}{2M} \quad \wedge \quad \mathrm{Pr}_{X \sim \hat{q}_{\mathrm{ERM}}}\left( \hat{v}(X) \neq v(X) \right) \leq \frac{\gamma \epsilon_2}{2}.$$

By a union bound, with probability $\geq 1 - \delta$, all of these estimation accuracy events take place. We condition on these favorable events taking place going forwards.

Note that conditioned on these favorable events, the normalizing constant $\hat{q}_{\mathrm{ERM}}\left( \{\hat{v}(x) = 1\} \right) > 0$, as for any ERM, we have

$$\hat{q}_{\mathrm{ERM}}\left( \{\hat{v}(x) = 1\} \right) = \mathbb{E}_{X \sim \hat{q}_{\mathrm{ERM}}}\left[ \mathbb{1}[v(x) = 1] \right] - \int \left( \mathbb{1}[v(x) = 1] - \mathbb{1}[\hat{v}(x) = 1] \right) d\hat{q}_{\mathrm{ERM}}(x)$$

$$\geq \mathbb{E}_{X \sim \hat{q}_{\mathrm{ERM}}}\left[ \mathbb{1}[v(x) = 1] \right] - \int \mathbb{1}[v(x) \neq \hat{v}(x)] d\hat{q}_{\mathrm{ERM}}(x)$$

$$\geq \gamma - \frac{\gamma \epsilon_2}{2}$$

$$> 0.$$

Thus, the restriction of the ERM to the estimated validity region is a viable probability distribution, and is outputted by the algorithm as $\hat{q}$. For any estimate of the validity function $\hat{v}$, we can decompose the loss of $\hat{q}$ as

$$L_P(\hat{q}; l) = \mathbb{E}_{X \sim P}\left[ l\left(f_{\hat{q}}(X)\right) \cdot \mathbb{1}[\hat{v}(X) = v(X)] \right] + \mathbb{E}_{X \sim P}\left[ l\left(f_{\hat{q}}(X)\right) \cdot \mathbb{1}[\hat{v}(X) \neq v(X)] \right].$$

First using Lemma 6, and then using the uniform convergence of the loss estimates, we can bound the first term as

$$\mathbb{E}_{X \sim P}\left[ l\left(f_{\hat{q}}(X)\right) \cdot \mathbb{1}[\hat{v}(X) = v(X)] \right] \leq L_P(\hat{q}_{\mathrm{ERM}}; l)$$

$$\leq L_P(q_l^*; l) + \frac{\epsilon_1}{2}$$

$$\leq L_P(q^*; l) + \frac{\epsilon_1}{2},$$

where $q_l^* = \arg\min_{q \in \mathcal{Q}} L_P(q; l)$ is the lowest-loss model in the class $\mathcal{Q}$. To upper bound the second term in the loss decomposition, we can use the fact that $\mathrm{Pr}_{X \sim P}\left( \hat{v}(X) \neq v(X) \right) \leq \epsilon_1/2M$ and the upper bound on the loss to write

$$\mathbb{E}_{X \sim P}\left[ l\left(f_{\hat{q}}(X)\right) \cdot \mathbb{1}[\hat{v}(X) \neq v(X)] \right] \leq M \cdot \mathbb{E}_{X \sim P}\left[ \mathbb{1}[\hat{v}(X) \neq v(X)] \right] \leq \frac{\epsilon_1}{2},$$

---
**Algorithm 3** Restriction to ERM under Log-Loss without Validity Assumption
---
1: **procedure** VALID_RESTRICTION_LOG(Distribution Class $\mathcal{Q}$, Validity Class $\mathcal{V}$, $\mathcal{D}$, $\delta$, $\epsilon_1$, $\epsilon_2$)

2: $\quad$ $S \leftarrow \Omega\left(\frac{M^2(\log(|\mathcal{Q}|) + \log(1/\delta))}{\epsilon_1^2}\right)$ i.i.d samples $\sim P$

3: $\quad$ $\hat{q}_{\text{ERM}} \leftarrow \arg\min_{q \in \mathcal{Q}} \sum_{x \in S} \min(M, -\log(f_q(x)))$

4: $\quad$ $\tilde{q}_{\text{ERM}} \leftarrow (1 - \epsilon_1/8) \cdot \hat{q}_{\text{ERM}} + \epsilon_1/8 \cdot \mathcal{D}$ $\qquad\qquad$ $\triangleright$ Mix with constant validity $\mathcal{D}$

5: $\quad$ $S_P \leftarrow \Omega\left(\frac{M(D\log(M/\epsilon_1) + \log(1/\delta))}{\epsilon_1}\right)$ i.i.d. samples $\sim P$,

$\qquad\qquad$ $S_{\tilde{q}_{\text{ERM}}} \leftarrow \Omega\left(\frac{D\log(1/\epsilon_1\epsilon_2) + \log(1/\delta)}{\epsilon_1\epsilon_2}\right)$ i.i.d. samples $\sim \tilde{q}_{\text{ERM}}$

6: $\quad$ $\hat{v} \leftarrow \arg\min_{h \in \mathcal{V}} \sum_{x \in S_P \cup S_{\tilde{q}_{\text{ERM}}}} \mathbb{1}[h(x) \neq v(x)]$ $\qquad\qquad$ $\triangleright$ Label $x \in S_{\tilde{q}_{\text{ERM}}}$ via $v$

7: $\quad$ **return** $f_{\hat{q}} \propto f_{\tilde{q}_{\text{ERM}}}(x) \cdot \mathbb{1}[\hat{v}(x) = 1]$ **if** $\tilde{q}_{\text{ERM}}(\{\hat{v}(x) = 1\}) > 0$ **else** $f_{\hat{q}} = f_{\hat{q}_{\text{ERM}}}$

8: **end procedure**
---

yielding the loss guarantee.

The validity guarantee follows directly from the fact that $\Pr_{X \sim \hat{q}_{\text{ERM}}}(\hat{v}(X) \neq v(X)) \leq \gamma\epsilon_2/2$ and Lemma 5, where $\gamma$ furnishes the lower estimate for the validity of the model $\hat{q}_{\text{ERM}}$. $\qquad\square$

The corollary to Theorem 3 stating that only low-loss models need appreciable validity is straightforwards. One can simply add an extra line to the proof of Theorem 3, arguing that when the intersection of good estimation events takes place, the loss of the ERM distribution is within $O(\epsilon_1)$ of the optimal loss across models in $\mathcal{Q}$, meaning that it has validity greater than some constant $c$. Thus, one can run Algorithm 2 with an $S_{\hat{q}_{\text{ERM}}}$ large enough to achieve $O(\epsilon_2)$ disagreement rate between $\hat{v}$ and $v$ under samples from $\hat{q}_{\text{ERM}}$, lowering the label complexity.

The proof of Theorem 4 is very similar to that of Theorem 3. The main difference is that when the loss is the capped log-loss, we can exploit a stability property under mixture similar to that introduced in Lemma 4. This allows us to mix $\hat{q}_{\text{ERM}}$ with a distribution of constant validity to get a validity lower bound on the final proposal distribution $\tilde{q}_{\text{ERM}}$ without increasing the loss more than $O(\epsilon_1)$. The validity lower bound can then be used as in Theorem 3.

**Theorem 4.** *Suppose $v \in \mathcal{V}$ where $VC(\mathcal{V}) \leq D$, and that for each $q \in \mathcal{Q}$, we have $f_q(x) \leq \beta$. Suppose further that there is some known $\mathcal{D} \in \mathcal{P}$ with density $f_{\mathcal{D}}$ which has $V(\mathcal{D}) \geq c > 0$ for some constant $c$. Then for all choices of $0 < \epsilon_1, \epsilon_2, \delta < 1/2$ and $M > 0$, Algorithm 3 requires a number of samples*

$$\leq \tilde{O}\left(\frac{M^2(\log(|\mathcal{Q}|) + \log(1/\delta))}{\epsilon_1^2} + \frac{M(D\log(M/\epsilon_1) + \log(1/\delta))}{\epsilon_1}\right),$$

*and a number of validity queries*

$$\leq O\left(\frac{D\log(1/\epsilon_1\epsilon_2) + \log(1/\delta)}{\epsilon_1\epsilon_2}\right),$$

*to ensure that with probability $\geq 1 - \delta$, its output enjoys*

$$\mathbb{E}_{X \sim P}\left[\min(M, \log(1/f_{\hat{q}}(X)))\right] \leq \mathbb{E}_{X \sim P}\left[\min(M, \log(1/f_{q^*}(X)))\right] + \epsilon_1 \quad and \quad I(\hat{q}) \leq \epsilon_2.$$

*Proof.* WLOG assume $\beta \geq 1$, and consider learning over $\bar{l}(z) = \min(M, \log(1/z)) - \log(1/\beta)$, a translation of the capped log-loss bounded below by 0 for all inputs to $f_q \in \mathcal{Q}$, and bounded above by $\bar{M} = M - \log(1/\beta)$.

Similar to the proof of Theorem 3, with probability $\geq 1 - \delta/2$ over the sample $S \sim P^n$, it holds that for all $q \in \mathcal{Q}$ that $\left|L_S(q; \bar{l}) - L_P(q; \bar{l})\right| \leq \epsilon_1/8$; in this case, we use the fact that $f_q \leq \beta$ to ensure that the random variables $\bar{l}(f_q(X))$ for $X \sim P$ are bounded, allowing for an application of

Hoeffding's inequality over empirical estimates of a loss unbounded below. As above, for any choice of $\tilde{q}_{\text{ERM}}$, the sizes of $S_P$ and $S_{\tilde{q}_{\text{ERM}}}$ are such that with probability $\geq 1 - 2\delta/3$,

$$\Pr_{X \sim P}\left(\hat{v}(X) \neq v(X)\right) \leq \frac{\epsilon_1}{2M} \quad \wedge \quad \Pr_{X \sim \tilde{q}_{\text{ERM}}}\left(\hat{v}(X) \neq v(X)\right) \leq \frac{c\epsilon_1\epsilon_2}{16}.$$

As before, by a union bound, these bounds both hold simultaneously with probability $\geq 1 - \delta$. We condition on this intersection of favorable events going forwards.

Note that when this intersection of events takes place, we have $\tilde{q}_{\text{ERM}}\left(\{\hat{v}(x) = 1\}\right) > 0$. In this case, we have that $V(\tilde{q}_{\text{ERM}}) \geq \epsilon_1 V(\mathcal{D})/8 \geq \epsilon_1 c/8$ as $V(\mathcal{D}) \geq c$, and so identically to our work in Theorem 3, we may write

$$\tilde{q}_{\text{ERM}}\left(\{\hat{v}(x) = 1\}\right) \geq \mathbb{E}_{X \sim \tilde{q}_{\text{ERM}}}\left[\mathbb{1}[v(X) = 1]\right] - \int \mathbb{1}[\hat{v}(x) \neq v(x)]d\tilde{q}_{\text{ERM}}(x)$$

$$\geq \frac{c\epsilon_1}{8} - \frac{c\epsilon_1\epsilon_2}{16}$$

$$> 0.$$

Thus, the restriction of $\tilde{q}_{\text{ERM}}$ to the estimate of the valid region is defined and outputted by the algorithm as $\hat{q}$. To see that the loss guarantee then holds for such a $\hat{q}$, consider the loss decomposition used in the proof of Theorem 3:

$$L_P(\hat{q}; \bar{l}) = \mathbb{E}_{X \sim P}\left[\bar{l}\left(f_{\hat{q}}(X)\right)\mathbb{1}[\hat{v}(X) = v(X)]\right] + \mathbb{E}_{X \sim P}\left[\bar{l}\left(f_{\hat{q}}(X)\right)\mathbb{1}[\hat{v}(X) \neq v(X)]\right].$$

We upper bound the second term exactly as in Theorem 3. To upper bound the first term, consider an argument similar to that of the proof of Lemma 6. Let $V_{\hat{v}}(\tilde{q}_{\text{ERM}}) > 0$ be the normalizing constant for $\hat{q}$. We can write

$$\mathbb{E}_{X \sim P}\left[\bar{l}\left(f_{\hat{q}}(X)\right)\mathbb{1}[\hat{v}(X) = v(X)]\right] \leq \mathbb{E}_{X \sim P}\left[\bar{l}\left(\frac{f_{\tilde{q}_{\text{ERM}}}(X)\mathbb{1}[\hat{v}(X) = 1]}{V_{\hat{v}}(\tilde{q}_{\text{ERM}})}\right) \cdot \mathbb{1}[\hat{v}(X) = 1]\right]$$

$$\leq \mathbb{E}_{X \sim P}\left[\bar{l}\left(\frac{f_{\tilde{q}_{\text{ERM}}}(X)}{V_{\hat{v}}(\tilde{q}_{\text{ERM}})}\right)\right]$$

$$\leq \mathbb{E}_{X \sim P}\left[\bar{l}\left(f_{\tilde{q}_{\text{ERM}}}(X)\right)\right].$$

Here, the non-increasingness of the loss still holds, leading to the final step. Now, fix some $x \in \mathcal{X}$ arbitrarily, and note the following, in the style of Lemma 4:

$$\bar{l}(f_{\tilde{q}_{\text{ERM}}}(x)) + \log(1/\beta) = \log\left(1/f_{\tilde{q}_{\text{ERM}}}(x)\right) \wedge M$$

$$\leq \left(\log\left(\frac{1}{1 - \epsilon_1/8}\right) + \log\left(1/f_{\hat{q}_{\text{ERM}}}(x)\right)\right) \wedge M$$

$$\leq \left(\epsilon_1/4 + \log\left(1/f_{\hat{q}_{\text{ERM}}}(x)\right)\right) \wedge M$$

$$\leq \log\left(1/f_{\hat{q}_{\text{ERM}}}(x)\right) \wedge M + \frac{\epsilon_1}{4}.$$

Thus, it holds that $\bar{l}(f_{\tilde{q}_{\text{ERM}}}(x)) \leq \bar{l}(f_{\hat{q}_{\text{ERM}}}(x)) + \epsilon_1/4$, and so we may write

$$\mathbb{E}_{X \sim P}\left[\bar{l}(f_{\hat{q}}(X))\right] \leq \mathbb{E}_{X \sim P}\left[\bar{l}\left(f_{\hat{q}_{\text{ERM}}}(X)\right)\right] + \frac{\epsilon_1}{4}$$

$$= L_P(\hat{q}_{\text{ERM}}; \bar{l}) + \frac{\epsilon_1}{4}.$$

Thus, we have written the loss in terms of the ERM, and so we have, as in Theorem 3, that the first term of the loss decomposition can be bounded by $L_P(q^*) + \epsilon_1/2$.

Given $\Pr_{X \sim \tilde{q}_{\text{ERM}}}\left(\hat{v}(X) \neq v(X)\right) \leq c\epsilon_1\epsilon_2/16$, and the lower bound on the validity of $\tilde{q}_{\text{ERM}}$ derived in the third paragraph above, we can again apply Lemma 5 to get the validity guarantee. $\square$

