# OpenReview forum: "Distribution Learning with Valid Outputs Beyond the Worst-Case"
_NeurIPS.cc/2024/Conference — NeurIPS 2024 poster_

### Official Review · Reviewer_KhXL · 2024-06-12

**Soundness:** 4
**Presentation:** 4
**Contribution:** 4
**Rating:** 8
**Confidence:** 4

**Summary:**

This paper follows up on the line of study initiated in the work "Actively Avoiding Nonsense in Generative Models" by Hanneke et al. 2018. The model studied in that paper is as follows: we are given data generated by a distribution $P$ over a domain $X$. A certain fraction of the domain $X$ is labeled "valid" and the rest is "invalid"; $P$ is only supported over valid samples. We are given access to an oracle that tells us whether any point $x \in X$ is valid or not. Our goal is to obtain a distribution $\hat{q}$, which minimizes a certain loss, subject to not having too much mass on the invalid fraction of the domain. More precisely, our benchmark is against a specified family of distributions $Q$. Let $q^\star$ be the distribution that minimizes a loss function among all the distributions $q \in Q$ that are fully supported on valid data. The distribution $\hat{q}$ we return should suffer an excess loss of at most $\epsilon_1$ compared to $q^\star$. Furthermore, the mass that $\hat{q}$ assigns to invalid samples should be at most $\epsilon_2$. To obtain such a $\hat{q}$, we can do 2 actions: 1) obtain iid samples from the data distribution $P$ 2) issue queries to the validity oracle. To obtain the required $\hat{q}$, we ideally want both our sample complexity as well as query complexity to be polynomial in $1/\epsilon_1$, $1/\epsilon_2$ and possibly also in the range of the loss function (i.e., if it is bounded in $[0,M]$ for $M < \infty$).


Among other results, the paper by Hanneke et al. 2018 shows that 1) If the learning algorithm is proper (constrained to return a distribution in $Q$), then even with infinite running time and infinite samples from $P$, it must issue at least $2^{\Omega(1/\epsilon_1)}$ queries to the validity oracle. 2) If the learning algorithm is improper, it can return $\hat{q}$ satisfying the required criteria with $O(M^2 \log |Q|/\epsilon_1^2)$ samples from $P$ and $O(M^2\log|Q|/\epsilon_1^2\epsilon_2)$ queries to the validity oracle. These results are stated for a very general class of loss functions (just bounded, monotonic decreasing).


The main contribution of this work is to study under what specialized settings the guarantees of Hanneke et al. 2018 can be improved, especially in the number of validity queries that learning algorithms require. The authors study two different scenarios, and obtain improved query complexities for each.


First, the authors consider a setting where 1) the true data distribution $P$ also belongs to the benchmark class $Q$ 2) the loss function is simply the log-loss function i.e. $l(x)=\log(1/\hat{q}(x))$ (where we are abusing notation so that $\hat{q}(x)$ is the density that $\hat{q}$ assigns to $x$). The former is a "realizability" condition, and the latter is natural since in practice, a default training choice is to maximize the likelihood of the observed data, which is equivalent to minimizing the log-loss. For this setting, the authors present an algorithm that obtains the desired $\hat{q}$ with **no queries to the validity oracle at all**---but now, the algorithm requires $\tilde{O}(\log|Q|/\min(\epsilon_1^2, \epsilon_2))$ samples, as compared to the $O(M^2\log|Q|/\epsilon_1^2)$ samples required by the more general improper algorithm of Hanneke et al. 2018. (Note that the log-loss is not bounded at all, and it helps to think of the comparison when $M$ is a constant). The learning algorithm simply returns the distribution in $Q$ that minimizes the emipirical loss, but suitably mixed with the uniform distribution over the domain $X$. Thus, under the assumptions of realizability, and with a specific loss function, the punchline is that validity comes easily from random examples themselves. The analysis uses classic tools from hypothesis testing like the Neyman-Pearson lemma (this is where the realizability assumption as well as the log-loss shows up). The authors also argue that the sample complexity dependence on $\epsilon_2$ is more-or-less optimal---any proper learner must necessarily use $1/\epsilon_2$ samples (although it still might be possible that improper learners, like the one the authors produce, can do better).

Second, the authors return to the more general setting considered by Hanneke et al. 2018, i.e., $P$ need not belong to $Q$, and the loss function is a monotone decreasing function bounded in $[0,M]$. However, now the crucial assumption is that validity oracle, which is a function that maps $X$ to {0,1}, belongs to a function class $V$ of bounded VC dimension $d$. Here again, the authors obtain a natural algorithm that uses fewer validity queries than the one by Hanneke et al. 2018. First, the algorithm obtains a distribution from $Q$ that minimizes empirical loss. Then, the algorithm obtains extra samples from $P$ (in fact, from $P$ mixed with $\hat{q}$), and invokes the validity oracle on this sample, to obtain valid/invalid labels for all the examples so obtained. Then, the algorithm finds a function $\hat{v}$ in $V$ that agrees with this labeling as much as possible. Finally, $\hat{q}$ obtained in the previous step is re-normalized to only have mass on points that are rendered valid by $\hat{v}$. The authors show that this algorithm works (when all the distributions in $Q$ have at least a constant mass on valid examples) with a sample complexity of $O(M^2\log|Q|/\epsilon_1^2)$ and query complexity of $\tilde{O}(d/\min(\epsilon_1,\epsilon_2))$. Note that the latter is still an improvement (in the dependence on $\epsilon_1, \epsilon_2$). The authors are also able to prove a result without the assumption of constant valid mass, albeit with a slightly worse query complexity (that is still better than that of Hanneke et al. 2018) and which only works for the capped-log-loss.
Finally, the authors comment on ways to improve the query complexities of their results in this setting.

**Strengths:**

The authors present an interesting, suitably exhaustive and strong set of results for natural "special" cases of the problem of validity-constrained distribution learning. Both the settings that the authors study (realizability with log-loss, as well as validity oracle in a bounded VC class) in the paper are natural beyond-worst case instances of the problem, and may well form a reasonable model of practical scenarios. The gains in the realizable setting are particularly impressive---it is nice to see that one can in principle do away with the validity oracle, while also maintaining near-optimal sample complexity. Even in the second setting, which just adds the additional assumption of a bounded VC validity region over those of Hanneke et al. 2018, the authors obtain improved query complexities. Overall, the suite of results feels satisfying and paints a meaningful beyond-worst case picture of the problem.

**Weaknesses:**

While I do find the study compelling, if I am to nitpick and search for weaknesses, I will say that there are no proof sketches whatsoever for the main theorems in the body. Notably, the content is still well under the page limit, and hence I suggest that the authors at least attempt to summarize the main steps used in the proofs of their results in the main body. Also, certain points and sentences could have used more elaboration and verbosity, to make the reading slightly less heavy. For example, lines 251-252 were not coherent to me and seemed like a mouthful, and similarly, at a few other places, I felt like more exposition could have been useful for the reader.

**Questions:**

Here are a few questions that came to my mind as I was reading (some may not be well-formed/basic misunderstandings):

1) In the guarantee of Theorem 1, isn't $q^\star= P$? maybe it would help to write a remark saying this.
2) From a skim, the exponential lower bound for proper learners in Hanneke et al. 2018 uses the coverage loss function. Do you think that such a bound for proper learners might also hold under non-realizability with the log-loss? Basically, I am trying to see if the reason for the exponentially many samples can be isolated to non-realizability alone, or if it also requires the unnatural loss function
3) In lines 250-252, what do you mean by the "realizable complexity"? Shouldn't this be "proper sample complexity" instead? Isn't it possible that there exists an improper learner in the realizable setting that gets something smaller than $1/\epsilon_2$?
4) Just to clarify, in Algorithm 2, in line 4, is it true that all the samples obtained in the previous line from $P$ will be labeled as valid by the validity oracle? I this only really saves a constant factor, but still, these samples need not be queried to the oracle.
5) From a skim of the proofs in the Appendix, it seems that the log-loss is only used to the degree that it is equivalent to likelihood maximization (e.g. in the proof of Lemma 6). If that is the case, can you say something more generic, that your results also hold for any loss functions that get minimized when the likelihood of the observed data gets maximized? If this is not the case, could you point out where exactly the exact form of the log-loss (i.e. $l(x)=\log(1/q(x))$ is crucial in your arguments?
6) I may be wrong and misunderstanding the measure notation, but it seems to me that in the equation block at line 470, in the second line, the $f^n_P(x)$ shouldn't be there. This is because, at least in the discrete case with pmfs, $\sum_x \min(p(x), q(x)) = 1-TV(p,q)$ (note that there is no $p(x)$ multiplying the $\min$ in the summation).
7) Could you explain a little more as to why in Algorithm 2, one needs to obtain samples from the mixture of $P$ with $\hat{q}$? Why do the samples from $\hat{q}$ end up being necessary?
---
Minor typos:

Line 112: typo any \
Line 136 any choice of \
Line 142 samples \
Line 249 means of \
Line 250-252 is not coherent and can use some elaboration \
Line 286 $d$ should be $D$

**Limitations:**

The authors have adequately addressed limitations as far as I can see.

---

> ### Author Rebuttal · Authors · 2024-08-05
>
> We thank the reviewer for taking the time to read our work with such attentiveness, and write such a detailed review. We will work to clarify the writing in the next version, and agree that some proof intuition can be added.
>
> Answers to the reviewer's questions can be found below:
>
> 1) This is correct: in the realizable case $q^* = P$. Theorem 1 was phrased in terms of $q^*$ mostly for notational consistency, but we should note the confluence here.
>
> 2) While we do not have a formal result, our impression is that the driving issue here is not the coverage loss, but the sort of extreme case of non-realizability they construct. When the supports of the distributions in the model class $\mathcal{Q}$ have so little overlap with the data generating distribution $P$, there is no way to ascertain which models are valid and which aren't without something close to exhaustively searching through them. The choice of loss does not feel helpful in resolving this -- for example, there is nothing about the maximum likelihood model $q$ that guarantees any sort of validity in the worst case.
>
> 3) Our writing in this paragraph is imprecise, and will be cleaned in the next version. By "realizable complexity" we were referring to the fact that $P \in \mathcal{Q}$ in the setting of Theorem 1. Given that the lower bound only applies to proper learners, the reviewer is correct in saying that it's possible there is an improper algorithm that surpasses the $\Omega(1/\epsilon_2)$ lower bound. That said, this feels to us somewhat unlikely given that the improperness of Algorithm 1 arises so that we can control the loss -- it is done so that we don't output a hypothesis that is small in total variation to $P$ but large in KL-divergence. From the perspective of invalidity, small total variation to $P$ is sufficient.
>
> 4) The reviewer is correct: the samples from $P$ do not need to be queried for validity, and updating the algorithm to reflect this saves a constant factor. We left this detail out originally, but will add it to Algorithms 2 and 3 in the next version.
>
> 5) The reviewer is correct that likelihood maximization is really what's doing most of the legwork in Theorem 1 (via Lemma 6). There is one other feature of the log-loss (see Lemma 8) which we use in Theorem 1 and Theorem 4: for mixture distributions $f_M(x) = (1-\epsilon) f_q(x) + \epsilon f_{r}(x)$, the form of the log-loss admits the inequality $\log(1/f_M(x)) \leq 2\epsilon + \log(1/f_q(x))$. In the case of Theorem 1, this allows us to argue that mixing with the uniform distribution does not degrade the loss too much. There are definitely some nice generalizations that can be made.
>
> 6) This is a good catch. The $f^n_P$ outside of the minimum in line 470 is a typo. The proof is correct when it is removed.
>
> 7) We use samples from both $P$ and the ERM in Algorithm 2 is so that we get a single estimate of the validity function $\hat{v}$ which has a high probability of agreement with the true validity function $v$ under samples from both distributions. Basically, we use the accuracy of the estimate $\hat{v}$ under $P$ to ensure that the loss of the returned distribution is small, and the accuracy of the estimate $\hat{v}$ under the ERM is to bound the invalidity. If we could return a distribution constructed by "accepting/rejecting" samples from the ERM using the true validity function $v$, we'd have perfect validity, but under an estimate of $v$, we need to make sure this estimate doesn't disagree too much with $v$ precisely under the proposal distribution $\hat{q}_{ERM}$, or too many invalid samples may be leaked.

---

> > ### Comment · Reviewer_KhXL · 2024-08-07
> > **Response to rebuttal**
> >
> > Thank you for the response. Please do revise the submission to reflect the discussion above. I maintain my score of 8, and believe this work should be accepted. Great job!

---

### Official Review · Reviewer_CS1K · 2024-07-16

**Soundness:** 3
**Presentation:** 3
**Contribution:** 3
**Rating:** 6
**Confidence:** 2

**Summary:**

The paper proposes a method for vastly improved validation query efficiency in validity-constrained distribution learning by relaxing the requirement for worst case settings of distributions, loss function and true validity function.

**Strengths:**

The paper makes good high level arguments for why regarding common case instead of worst case settings can be appropriate in some situations.

The proposed algorithm makes sense and are supported by mathematical derivations of its low error and high validity guarantees given assumptions on the space of learnable distributions.

The theorems look valid, though I didn't check the proofs.

**Weaknesses:**

A more in depth discussion of situations where one could clearly afford the proposed relaxation versus those where it might not be prudent enough would be helpful for justifying it.

**Questions:**

-

**Limitations:**

Yes.

---

> ### Author Rebuttal · Authors · 2024-08-05
>
> We appreciate the review, and will try to address the weaknesses they point out.
>
> We would argue that most practical situations are likely closer to the regime of realizability
> explored in the first half of this paper than the case where no model in the model class is a reasonable approximation for the data generating distribution, the setting which motivates algorithms in previous work [1].
>
> It also seems that valid regions of space are "learnable" in practice, motivating the second half of the paper where the validity function is known to lie in a bounded complexity class. For example, [3] found empirical success on the problem of post-editing GANs by coaxing GANs towards their restriction to "valid" parts of space.
>
> [1] Hanneke, S., Kalai, A., Kamath, G., & Tzamos, C. (2018). Actively Avoiding Nonsense in Generative Models. https://arxiv.org/abs/1810.06988 \
> [3] Kong, Z., & Chaudhuri, K. (2022). Data Redaction from Pre-Trained GANs. https://arxiv.org/abs/2206.14389

---

> > ### Comment · Reviewer_CS1K · 2024-08-12
> >
> > Thanks for the response!
> >
> > The point I mentioned in the review is that in different practical situations, the assumption might or might not be valid. Even if it is valid in most situation, some guidelines or measurable prerequisites that make the assumption more likely to safely hold would be helpful. This is however a minor point for the overall paper, and I'm maintaining my score.

---

### Official Review · Reviewer_HLjo · 2024-07-17

**Soundness:** 3
**Presentation:** 2
**Contribution:** 2
**Rating:** 4
**Confidence:** 3

**Summary:**

The paper considers the problem of learning generative models under validity constraints. Specifically, given data from a distribution and an oracle that tells whether a particular datapoint is valid, the goal is to output a generative model from a hypothesis class Q whose output has a small loss epsilon_1 (compared to the best model in Q) while ensuring the output is valid with high probability (1 - epsilon_2).  The objective is to achieve this with minimal training samples and validity queries to the oracle.

Previous work on this problem showed a negative result:  exp(1/epsilon_1) validity queries are needed to properly learn the generative model. For improper learning, they show that O(log(Q)/(epsilon_1^2 epsilon_2)) validity queries suffice. This work aims to improve these results by making further assumptions.

For example, it is shown that if the true data-generating distribution P is assumed part of class Q, then a minor modification of empirical risk minimization can achieve the loss and validity requirements with O(log(Q)/min(epsilon_1^2, epsilon_2))  samples and no validity queries. Additional results are provided, assuming a bound on the VC dimension of the class to which the validity oracle belongs.

**Strengths:**

The paper makes concrete improvements to the bounds established in previous works (under certain assumptions).

**Weaknesses:**

While the problem of generative modeling under validity constraints has been introduced citing practical concerns, it is unclear if the results or algorithms in this paper add much to generative modeling in practice. Therefore, the significance and relevance of these results are unclear.

**Questions:**

Could the authors elaborate on any practical insights that can be derived from the algorithms and bounds presented in this work?

**Limitations:**

See the weaknesses section. It would be great to include a discussion of insights for practice.

---

> ### Author Rebuttal · Authors · 2024-08-05
>
> We appreciate the review, and agree that the next version of the paper should furnish more practical insight.
>
> We mainly see these results as a first attempt at studying the problem of learning a valid generative model from the opposite perspective of [1], which paints learning in this setting as a complex endeavor. In practice, generative models do often suffer from invalidity issues, leading to a variety of techniques for mitigating this issue, many of which are quite intuitive and feasible [2, 4, 5, 6]. We hope this work can spark further investigation into uncovering natural settings where practical algorithms can be effective.
>
> Lemma 2  -- which shows that validity naturally arises when the data distribution $P$ is in the model class $\mathcal{Q}$ -- is intended to shed light on what learning a valid generative model is like when the model class $\mathcal{Q}$ contains good approximations of the "validity-filtered" data distribution. While the rates in Lemma 2/Theorem 1 hold only for the realizable case, we would argue that this setting likely a more faithful representation of generative modeling in practice than the more general setting of [1], where much of the complexity seems to come from the possibility that all models in $\mathcal{Q}$ bare little to no resemblance to the data generating distribution $P$. Basically, our claim here is that learning the distribution well is really what needs to be done – if you’ve learned well in this setting, you’re unlikely to generate invalid outputs.
>
> In the second half of the paper, we consider the possibility of learning the validity function and restricting the ERM model to the learned "valid" part of space. One recent applied work [3] similarly considers post-editing GANs by "learning the data distribution restricted to the complement of the [valid part of space]". Explaining the success of such schemes that operate on the estimation of the validity function requires relaxations of the original learning model of [1], where the validity function cannot be estimated well with a polynomial number of queries.
>
> [1] Hanneke, S., Kalai, A., Kamath, G., & Tzamos, C. (2018). Actively Avoiding Nonsense in Generative Models. https://arxiv.org/abs/1810.06988 \
> [2] Kaneko, T., & Harada, T. (2021). Blur, Noise, and Compression Robust Generative Adversarial Networks. https://arxiv.org/abs/2003.07849 \
> [3] Kong, Z., & Chaudhuri, K. (2022). Data Redaction from Pre-Trained GANs. https://arxiv.org/abs/2206.14389 \
> [4] Schramowski, P., Brack, M., Deiseroth, B., & Kersting, K. (2023). Safe Latent Diffusion: Mitigating Inappropriate Degeneration in Diffusion Models. https://arxiv.org/abs/2211.05105 \
> [5] Malnick, S., Avidan, S., & Fried, O. (2023). Taming Normalizing Flows. https://arxiv.org/abs/2211.16488 \
> [6] Liu, J., Xu, J., & Wang, Y. (2022). Efficient Retrieval-Augmented Generation: An Empirical Study of Practical Techniques. https://arxiv.org/abs/2210.04610

---

> > ### Comment · Reviewer_HLjo · 2024-08-11
> >
> > I appreciate the authors' response. While I agree that this work adds nuance to the prior work by Hanneke et al., the high-level takeaways and the algorithms do not seem to contribute significantly beyond what is already established in the practice of generative modeling. Therefore, I will maintain my original score.

---

### Official Review · Reviewer_yKWM · 2024-07-17

**Soundness:** 3
**Presentation:** 2
**Contribution:** 3
**Rating:** 6
**Confidence:** 3

**Summary:**

This paper studies distribution learning with invalidity constraints. It assumes that the algorithm has access to an oracle that provides validity queries, and targets to reduce the total amount of queries to the oracle while achieving comparable learning guarantees to previous counterparts. By specifying the loss function and restricting the hypothesis class, the paper achieved improved sample/query complexity of $\tilde{O}(\log{\lvert Q\rvert}/\min(\epsilon_1^2,\epsilon_2))$ (from previous $\tilde{O}(\log\lvert Q\rvert/\epsilon_1^2\epsilon_2)$ of Hanneke et al. 2018). Additionally, by assuming that the validity function is from a VC-class with VC-dim $D$, then the query complexity can be further reduced to $\tilde{O}(D/\min{\epsilon_1,\epsilon_2})$.

**Strengths:**

This paper considers a nontrivial problem, learning a generative distribution with validity verification. It contributes to the literature of both generative networks and distribution learning. On the other hand, it improves the query efficiency of the learning algorithm by specifying or restricting the class of distribution, loss function, class of validity functions. The progress on improving the query efficiency, see summary, looks a bit incremental; however, requires delicate handling in the algorithmic design and analysis. The theoretical analysis looks sound to me.

**Weaknesses:**

There are some typos. Line 136: choice of. Line 237: "is true for .. for .." To name a few.

Some parts are a bit unclear, for example, Line 37: "while achieving polynomial bounds on the number of validity queries, uses relative large number of validity queries". It is unclear what the "relative large number" is comparing to.

Line 232-233: "attainable, at least improperly". Does this mean that the algorithm can output an improper hypothesis? Is this due to the mixture of the ERM with the uniform distribution? Then, is proper learning achievable?

In addition, the format for the references is not unified. Some references are missing the information for venues.

**Questions:**

Do all algorithms run in exponential time, i.e. computationally inefficient?

**Limitations:**

I have no concerns on the potential negative societal impact.

---

> ### Author Rebuttal · Authors · 2024-08-05
>
> We thank the reviewer for looking closely at our work, and apologize for sometimes being less clear than we should.
>
> In line 37 ("while achieving polynomial bounds on the number of validity queries, uses relative large number of validity queries''), what we were referring to is that the improper algorithm of Hanneke et al. makes $\tilde{O}(1/\epsilon_1^2 \epsilon_2)$ validity queries, which looks large relative to the quadratic dependence one is used to seeing. It's an open question what the right query complexity is for the general case discussed by Hanneke et al. [1]., so this sentence should probably be reworked.
>
> In line 232-233 ("guarantees for the log-loss are attainable, at least improperly"), the reviewer is correct in assuming that by invoking “improper” learning here, we are referring to the fact that we output a mixture of the ERM and the uniform distribution in Algorithm 1. Satisfying the loss requirement with a proper learner will in general require many more samples than $\tilde{O}(1/\min(\epsilon_1^2, \epsilon_2)$, as there may be some $\tilde{P} \in \mathcal{Q}$ which is very close in total variation to $P$, but has infinite KL-divergence from $P$.
>
> With regard to computational considerations, the algorithms are in general inefficient. For efficiency, one needs access to an efficient ERM routine over the distribution class $\mathcal{Q}$ -- in this sense, our work follows in the line of original work of Hanneke et al. [1]., who assume access to an efficient (constrained) ERM oracle for computational efficiency. In the cases of Algorithms 2 and 3, one also needs access to an efficient ERM routine over the VC class. It's true that many important VC classes do not admit efficient ERM.
>
> [1]  Hanneke, S., Kalai, A., Kamath, G., & Tzamos, C. (2018). Actively Avoiding Nonsense in Generative Models, https://arxiv.org/abs/1810.06988

---

> > ### Comment · Reviewer_yKWM · 2024-08-12
> > **Official Comment by Reviewer yKWM**
> >
> > Thank you for your response. It will be helpful to include the discussion in the revision.

---

### Decision · Program_Chairs · 2024-09-25

**Decision:**

Accept (poster)

**Comment:**

This work studies the problem of approximating a target distribution under the constraint that the estimated distribution should not assign too high weight on invalid regions. Though prior work gave worst-case analysis that implies exponential sample complexity, this paper considers a few special models under which polynomial sample complexity can be obtained.

Though there is certain concern on the practical implication, the theoretical results were found interesting and sound, and the special cases  appear reasonable path to attack the problem.